# Ribosome rescue factor PELOTA modulates translation start site choice for C/EBPα protein isoforms

Samantha G Fernandez[1], Lucas Ferguson[1,2], Nicholas T Ingolia[1,2]

**Translation initiation at alternative start sites can dynamically control the synthesis of two or more functionally distinct protein isoforms from a single mRNA. Alternate isoforms of the developmental transcription factor CCAAT/enhancer-binding protein α (C/EBPα) produced from different start sites exert opposing effects during myeloid cell development. This choice between alternative start sites depends on sequence features of the *CEBPA* transcript, including a regulatory uORF, but the molecular basis is not fully understood. Here, we identify the factors that affect C/EBPα isoform choice using a sensitive and quantitative two-color fluorescent reporter coupled with CRISPRi screening. Our screen uncovered a role of the ribosome rescue factor PELOTA (PELO) in promoting the expression of the longer C/EBPα isoform by directly removing inhibitory unrecycled ribosomes and through indirect effects mediated by the mechanistic target of rapamycin kinase. Our work uncovers further links between ribosome recycling and translation reinitiation that regulate a key transcription factor, with implications for normal hematopoiesis and leukemogenesis.**

## Introduction

The regulation of translation initiation shapes cellular proteomes in organisms ranging from bacteria to humans. In eukaryotes, translation preinitiation complexes typically scan unidirectionally from the 5′ end of mRNAs and initiate translation at the first AUG codon (1, 2). However, cis-regulatory sequences present in the 5′ UTR can alter translation start site choice (3). Nature has leveraged this flexibility in start site selection to encode multiple alternative protein isoforms on a single transcript (4, 5, 6, 7, 8, 9, 10, 11, 12, 13, 14, 15, 16 Preprint). Initiation at alternative start sites produces N-terminal extensions and truncations that can affect the localization, stability, and even function of the resulting proteins, in some cases yielding protein isoforms with opposing effects. Indeed, functionally distinct isoforms produced from alternative initiation have wide-ranging consequences from cellular differentiation (17, 18) and development (19, 20) to cell cycle regulation (21 Preprint) and innate immune signaling (12). The protein machinery that controls the choice between translation start sites and thus the synthesis of these alternative protein isoforms are not well understood, however.

Alternative start sites also initiate translation of short uORFs—ubiquitous regulatory elements present in roughly half of all mammalian transcript leaders (22, 23). The translation of uORFs interferes with ribosomes reaching downstream coding sequences (CDSes), thereby repressing their translation (22, 24, 25, 26, 27). Productive translation of uORF-containing mRNAs therefore requires that ribosomes bypass uORF start codons in a process called leaky scanning. Alternatively, recognition of uORF start codons can lead to reinitiation wherein ribosomes translate the uORF, but then remain bound to the mRNA because of partial recycling after termination and reinitiate at a second downstream ORF (24, 26, 28). Typically, after translation termination, the ribosome is split apart and the large (60S) subunit is released (29, 30). The P-site tRNA is then removed, and finally, the small (40S) subunit is released from the mRNA as well, to complete the recycling process (31, 32). Translation reinitiation occurs when recycling is incomplete, and either the whole ribosome or the small subunit remains bound to the mRNA and begins translation again at a different start codon.

Few translation factors are known to selectively influence reinitiation; the best characterized reinitiation factors are the density-regulated reinitiation and release factor (DENR) and multiple copies in T-cell lymphoma-1 (MCTS1) complex, first shown to favor reinitiation in *Drosophila melanogaster* (33). By facilitating small subunit recycling after uORF translation, these factors modulate the stress-inducible synthesis of the bZIP transcription factor, ATF4 (34, 35, 36, 37). Although defects in large subunit recycling can also permit aberrant reinitiation, this is mechanistically distinct from small subunit recycling and not associated with uORF regulation (38, 39). In addition, one ribosome translating a uORF can block a second initiation complex from scanning past the uORF, and this inhibitory effect is stronger when ribosomes stall on the uORF (24, 26, 40, 41, 42, 43, 44). This kind of ribosome queuing has also been shown to increase recognition of upstream start codons (45, 46, 47, 48). Thus, the interplay between scanning, uORF

[1]Department of Molecular and Cell Biology, University of California, Berkeley, CA, USA   [2]Center for Computational Biology and California Institute for Quantitative Biosciences, University of California, Berkeley, CA, USA

Correspondence: ingolia@berkeley.edu

translation, and reinitiation can provide complex, 5′ UTR-encoded regulation to control protein expression.

Translation of the critical bZIP developmental transcription factors encoded by *CEBPA* and *CEBPB* is likewise regulated by uORFs (18, 49). In contrast to the three uORFs of *ATF4*, the single uORF of *CEBPA* regulates the synthesis of two distinct, alternative translation isoforms from the same single-exon mRNA (17, 50). The longer isoform of CCAAT/enhancer-binding protein α (C/EBPα) includes the full N-terminal transactivation domain. In contrast, a shorter isoform of C/EBPα, which is initiated from an internal, canonical AUG start codon, produces a truncated isoform that retains the bZIP DNA-binding domain, but lacks most of the transactivation domain (Fig 1A) and thus acts in a dominant-negative manner by blocking long C/EBPα binding and transactivation (51). The ratio between these isoforms changes during normal development (52, 53) where they play distinct roles during myeloid and adipocyte lineage commitment (18, 54) and regulate different transcriptional targets (55). Mutations that reduce long isoform expression are frequently seen in acute myeloid leukemia (AML), including mutations in *CEBPA* itself that occur in 10–15% of these cancers (51, 56), emphasizing that the stoichiometry of these two isoforms must be maintained for proper differentiation.

Previous work has shown that the C/EBPα isoform balance is dependent on a short six-codon uORF that ends just seven nucleotides before the long isoform start codon (18). Translation of the long isoform is therefore mutually exclusive with uORF translation and occurs through leaky scanning past the suboptimal uORF start site followed by initiation at the next start codon. Synthesis of the short isoform, in contrast, depends on translation of the uORF followed by reinitiation that bypasses the long isoform start codon and instead occurs at a downstream, internal AUG codon (Fig 1B). The molecular processes that determine start codon choice on *CEBPA* and thus the ratiometric balance between these opposing isoforms have not been fully elucidated.

Here we characterize the *trans*-regulatory landscape governing translational control of the C/EBPα isoform ratio using a dual fluorescent reporter coupled with CRISPRi screening. We find that reinitiation factors, including DENR/MCTS1, influence start site selection on *CEBPA*, and uncover a surprising role for the ribosome rescue factor PELO (57, 58, 59, 60) during or after uORF termination. Reporter variants and ribosome profiling suggest that PELO enhances long C/EBPα isoform expression directly by recycling and removing post-termination ribosomes that occlude its expression, in addition to indirect effects mediated by mechanistic target of rapamycin (mTOR) activation. The accumulation of unrecycled ribosomes may provide another layer of repression and point of regulation on *CEBPA* and other uORF-containing transcripts.

## Results

### A fluorescent reporter measures *CEBPA* translation start site choice

To perform a forward genetic screen for factors that affect translational control of alternative C/EBPα isoforms, we developed a reporter that converted the choice between translation start sites on *CEBPA* into a ratiometric fluorescence signal that we could use for fluorescence-activated cell sorting (FACS). Because the shorter isoform is an N-terminal truncation of the longer isoform, we could not simply mark each isoform with its own fluorescent protein. Instead, we fused the fast-folding, red fluorescent protein mScarlet-I (61) to the shared C-terminus; both isoforms would contain mScarlet, and red fluorescence would serve as a proxy for total protein abundance. We introduced a green fluorescent reporter into the long isoform–specific N-terminal extension where it would report specifically on long isoform levels, allowing us to measure the relative abundance of the two translational isoforms by the fluorescence ratio (Fig 1C) and select for cells with altered isoform ratios by FACS:

$$\frac{green}{red} \propto \frac{long\ isoform}{long\ isoform + short\ isoform}$$

The endogenous N-terminal extension on the long isoform is substantially shorter than a fluorescent protein, and its mRNA sequence may contain regulatory information. To minimize disruptions to the organization and regulation of the transcript, we encoded only the short (16 amino acid) fragment of the split self-complementing green fluorescent protein mNeonGreen2 ($mNG2_{11}$) (62, 63) in the *CEBPA* CDS between the long and short isoform start sites. The coexpression of our long isoform reporter with the larger fragment of mNeonGreen2 ($mNG2_{1–10}$) reconstituted green fluorescence (Fig S1A). To increase the dynamic range and sensitivity of our reporter, we optimized the Kozak sequence around the uORF start codon to enhance short isoform expression (Table S1). We further deleted the DNA-binding domain of C/EBPα to mitigate any secondary transcriptional effects of our reporter. We then stably integrated a single copy of this construct into the *AAVS1* locus in K562 human myeloid leukemia cells, which do not endogenously express C/EBPα protein. In addition, we stably expressed $mNG2_{1–10}$ in these cell lines to generate monoclonal reporter cell lines (Fig S1B).

To verify that our reporter recapitulates the regulated choice between *CEBPA* translation start sites, we first tested its dependence on the start codons. Mutation of the uORF start codon abolished short isoform expression by eliminating reinitiation, and mutation of the long isoform start codon (Table S1) itself eliminated long isoform expression as evidenced by Western blot analysis using an anti-C/EBPα antibody (Fig 1D). In agreement with these results, loss of short isoform expression increased the ratio of green to red fluorescence, confirming that our fluorescence measurements accurately reflected the underlying protein isoform ratio (Fig 1E). We further tested how fluorescence of the WT reporter changed under conditions that shift C/EBPα isoform expression. Treatment with the allosteric mTOR inhibitor rapamycin reduces short isoform expression (18). We recapitulated this effect in our system by treating our reporter cell line with the mTOR active-site inhibitor PP242 (64, 65) and observed an increase in the ratio between green and red fluorescence, relative to DMSO-treated cells, consistent with a shift toward long isoform expression (Fig S1C). Thus, our fluorescent reporter measures changes in the

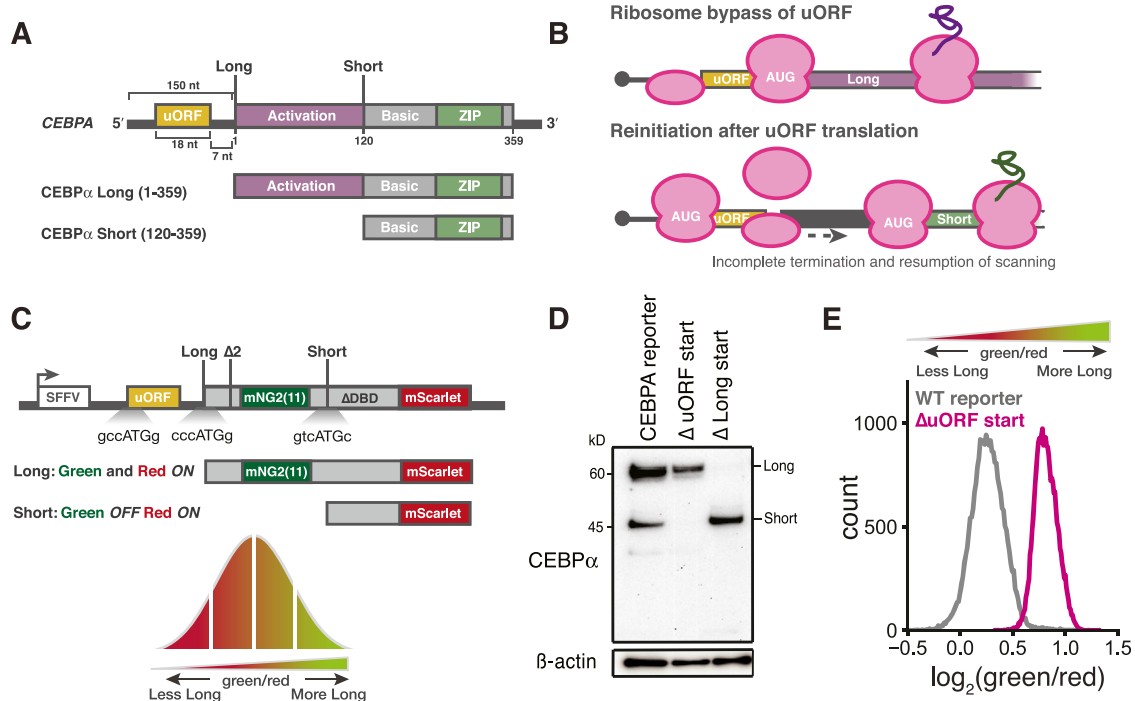

**Figure 1. Quantitative two-color fluorescent reporter for measuring start site selection on *CEBPA*.**
**(A)** Schematic representation of the *CEBPA* transcript (top) and protein isoforms (bottom). **(B)** Model of long isoform translation by leaky scanning and short isoform translation by reinitiation. **(C)** Schematic of the two-color reporter. Start codon context for the uORF, long and short start sites are indicated. Δ2: mutation of a second in-frame start codon; ΔDBD: deletion of the DNA-binding domain. **(D)** Western blot of K562 cells stably expressing CEBPα reporters. Δ uORF start: AUG→ACA; Δ Long start: AUG→ACA. **(E)** Flow cytometry measurements of the green/red fluorescence ratio in K562 cells stably expressing indicated reporters.

isoform ratio resulting from the choice between translation start sites.

## CRISPRi screens identify genes that modulate *CEBPA* start site choice

We then set out to identify factors that modulate *CEBPA* translation start site selection from a pooled library of CRISPR-based genetic perturbations using our fluorescent readout. Because many translation factors are essential and mutants can provoke strong growth defects, we perturbed gene expression by CRISPR interference (CRISPRi) (66), which produces strong partial loss-of-function phenotypes that allow uniform comparisons across essential and nonessential genes. We transduced our reporter cell line with four different lentiviral CRISPRi sgRNA sublibraries that collectively comprised 57,900 guides targeting ~10,000 genes, in addition to 1,070 nontargeting control guides (67). These sub-libraries were composed of guides classified under the following functional categories: "Gene Expression," "Stress and Proteostasis," "Cancer and Apoptosis," and "Kinases, Phosphatases, and Drug Targets." We carried out FACS to distribute transduced cells into four distinct bins depending on their green/red fluorescence ratio and quantified the relative frequency of each sgRNA across the four bins by high-throughput sequencing (Fig 2A); cells expressing a CRISPRi sgRNA that alters the isoform ratio should be unequally distributed across these four FACS bins (Fig S2A and B). Indeed, although sgRNA abundance generally correlated well between

adjacent bins (Spearman's ρ = 0.75 – 0.76) (Fig S2C and D), we identified dozens of sgRNAs with highly skewed distributions indicative of a change in the C/EBPα isoform ratio (Figs 2B–D and S2C). We quantified the shift in the isoform ratio for each sgRNA using a generalized linear model of its abundance in the four sorted bins (Fig 2C and D and Supplemental Data 1). In many cases, two or more independent sgRNAs targeting the same gene caused significant shifts, whereas the vast majority of our 1,070 nontargeting sgRNAs showed no significant effect, arguing that our approach was robust and specific (Figs 2C and D and S2E).

Many translation factors emerged among the targets with the strongest and most significant changes in the isoform ratio. These included sgRNAs against *DENR* and *MCTS1*, which greatly increased the green/red ratio, consistent with a defect in translation reinitiation and thus short isoform production (Figs 2B–D and S2C) (68, 69, 70, 71). Loss of *DENR* or *MCTS1* reduces reinitiation after uORF translation in flies (33), although work in human cells suggested that they primarily affect transcripts with extremely short single-codon uORFs (72). Depletion of other factors implicated in reinitiation, including the eIF4G paralog *eIF4G2/DAP5* (73, 74) and the eIF3 subunit *eIF3H* (75, 76), also increased the green/red ratio. More broadly, the targets that increased the short isoform fraction were enriched in gene ontology (GO) annotations for translation (Fig S2F and G), whereas targets that increased the long isoform fraction were enriched for GO annotations for nucleic acid binding and mRNA-related regulation (Fig 2E and F), although we also saw an effect of the post-translational regulator TRIB1 (77) (Fig S2H).

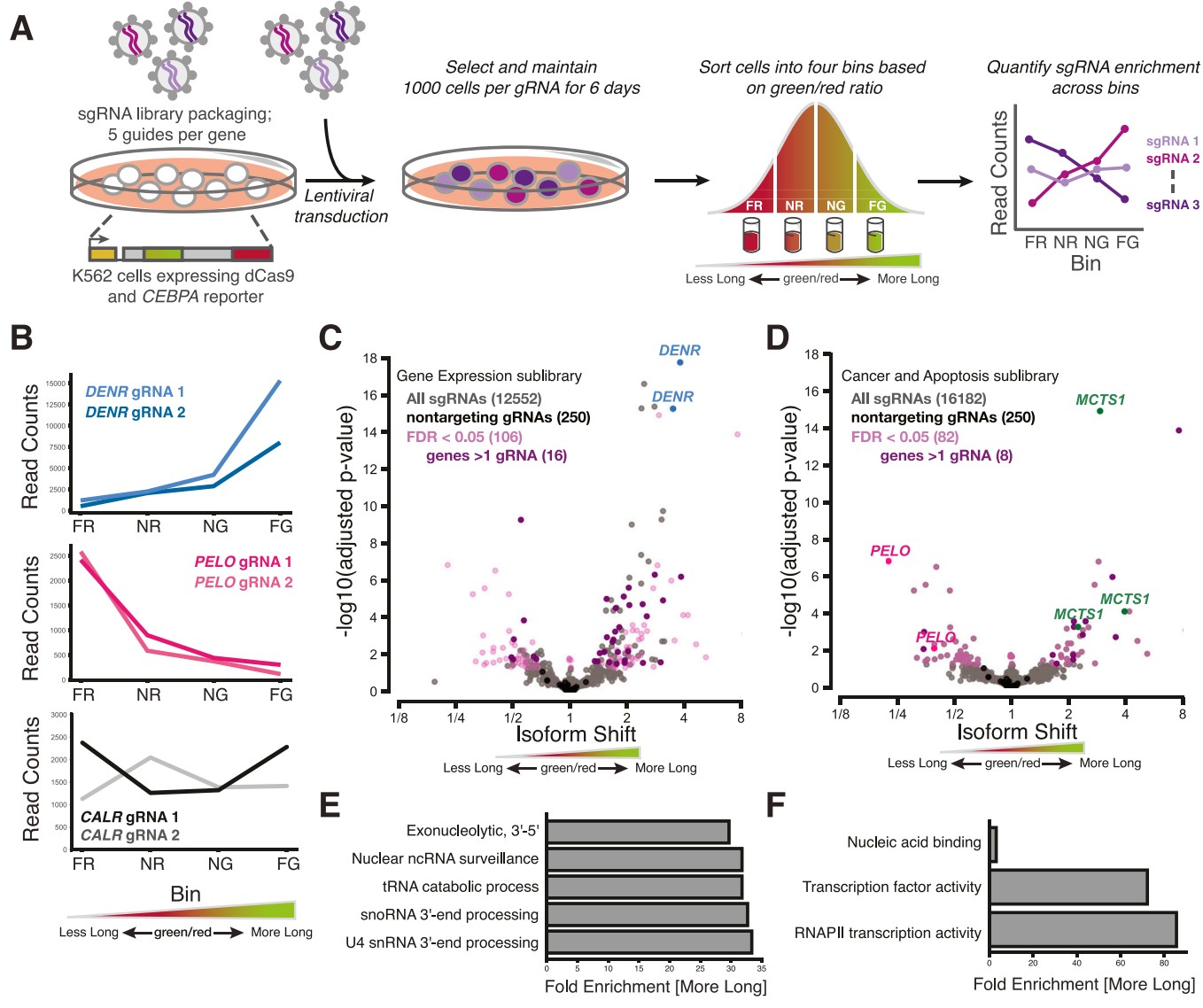

**Figure 2. CRISPRi sublibrary screens identify regulators of C/EBPα isoform expression.**
**(A)** Schematic of FACS-based CRISPRi screening strategy. Bin labels: FR, far red; NR, near red; NG, near green; FG, far green. **(B)** Distribution of sgRNA read counts across FACS bins for sgRNAs against *DENR*, *PELO*, and *CALR*. **(C)** "Gene Expression" sublibrary isoform shifts. Each point represents a single sgRNA with sgRNAs against *DENR* highlighted. **(D)** "Cancer and Apoptosis" sublibrary, as in (C). **(E)** Gene ontology (GO) terms associated with sgRNAs that were enriched in a significant shift towards the long isoform in the Gene Expression sublibrary screen (Fisher's exact test, Bonferroni's correction, fold enrichment >29). **(F)** As in (E), but with sgRNAs from the Cancer and Apoptosis sublibrary screen.

We also detected a decrease in the green/red ratio caused by depletion of the start codon stringency factors, *EIF1* and *EIF1A* (Fig S2G) (2, 78). It seems unlikely that reduced start codon stringency would increase uORF initiation because the reporter uORF Kozak sequence is optimized. Instead, this shift likely reflects other, potentially indirect, effects on the isoform ratio. Recent work has implicated DAP5 and eIF3 subunits in start codon stringency (79), and relaxed start codon selection may contribute to their effects. Although eIF5 plays an opposing role to eIF1 in start codon stringency, *EIF5* knockdown did not significantly change the isoform ratio, perhaps because robust autoregulation of eIF5 levels (80) renders it particularly resistant

to transcriptional inhibition. Interestingly, depletion of the initiation factor eIF2α, encoded by *EIF2S1*, actually decreased the green/red ratio, in contrast to the observation that increasing eIF2α availability causes a shift towards short isoform production (18); we speculate that extreme CRISPR-mediated eIF2α depletion may increase leaky scanning past the long isoform start codon as a result of reduced ternary complex formation (3). Consistent with this model, we also see that depleting subunits of eIF2B, which is required to regenerate ternary complex, cause a similar shift (Fig S2G).

Depletion of the ribosome rescue factor *Pelota* (PELO) strongly reduced the green/red ratio, indicating a major shift toward short

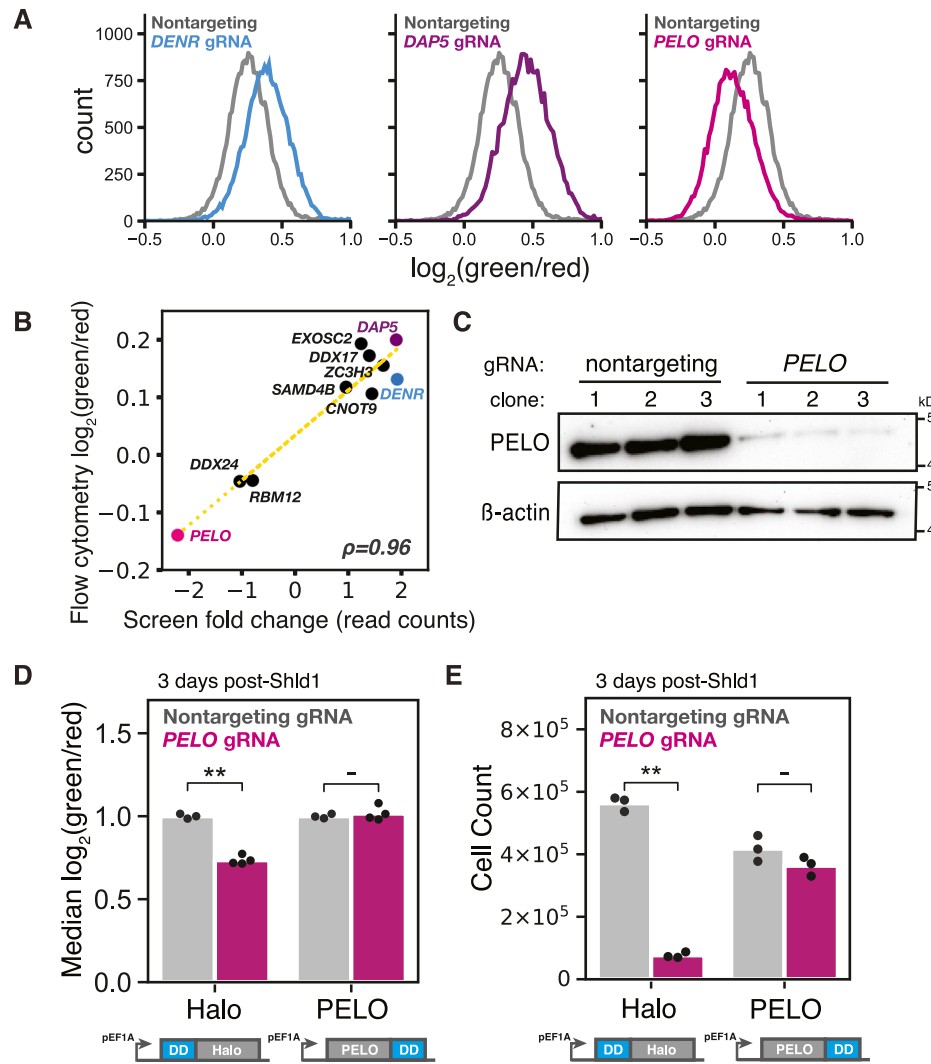

**Figure 3.  Validation of top CRISPRi sublibrary screen targets.**
**(A)** Flow cytometry measurements of cells transduced with indicated sgRNAs. **(B)** Comparison of isoform ratio shift in sublibrary screens and individual sgRNA knockdowns by flow cytometry, $\rho$ = 0.96. **(C)** Western blot of *PELO* knockdown by the top-scoring *PELO* sgRNA. Clones represent separate sgRNA transductions. **(D)** Flow cytometry measurements of isoform ratio rescue by PELO expression (n = 3 or 4; ** indicates $P < 0.01$, two-way ANOVA with Tukey's post-test). **(E)** Cell count measurements after PELO expression, as in (D) (n = 3).

isoform production (Figs 2B and D and S2C). PELO functions in translational quality control pathways to release stalled ribosomes from truncated transcripts and to remove unrecycled post-termination ribosomes from 3′ UTRs or at the ends of truncated transcripts (30, 57, 58, 59, 60, 81). Although PELO functions in translation and suppresses defects in large subunit recycling, it was not previously linked to uORF-mediated regulation, suggesting a distinct and perhaps *CEBPA*-specific role. PELO acts in conjunction with another recycling factor called HBS1L (HBS1 in yeast) to mediate ribosome recycling; all five sgRNAs against *HBS1L* shifted the ratio modestly yet similarly to *PELO* depletion, but none of their effects were significant. The lack of a strong effect from sgRNAs targeting *HBS1L* may be explained in part by observations that *HBS1L* knockout leads to a less dramatic translation phenotype than *PELO* knockout (82), and thus may have a correspondingly weaker effect on our reporter. Furthermore, many blood cells express alternate *HBS1L* transcripts from two distinct promoters separated by thousands of base pairs (83), and so we expect that CRISPRi knockdown of *HBS1L* will be inefficient. Interestingly, PELO

is homologous to the peptide release factor encoded by *ETF1* that acts in normal translation termination (84), and two sgRNAs targeting *ETF1* showed a similar but weaker shift toward short isoform translation (Fig S2G). These results suggested that termination and recycling—perhaps after uORF translation—could affect start site choice and thus C/EBPα isoform ratios.

### Reinitiation and ribosome rescue factors control *CEBPA* start site choice

We selected several genes with known roles in translation and RNA biology for individual validation. Two independent clonal cell lines expressing a sgRNA against *DENR* had a higher green/red fluorescence ratio than cells expressing a nontargeting control, in agreement with our results from flow sorting and sequencing; we saw similar results in two independent clones expressing sgRNAs against *DAP5*. We also recapitulated a lower green/red ratio in two clonal cell lines expressing a sgRNA against *PELO* (Figs 3A and S3A). More broadly, we saw a strong correlation (r = 0.96) between flow

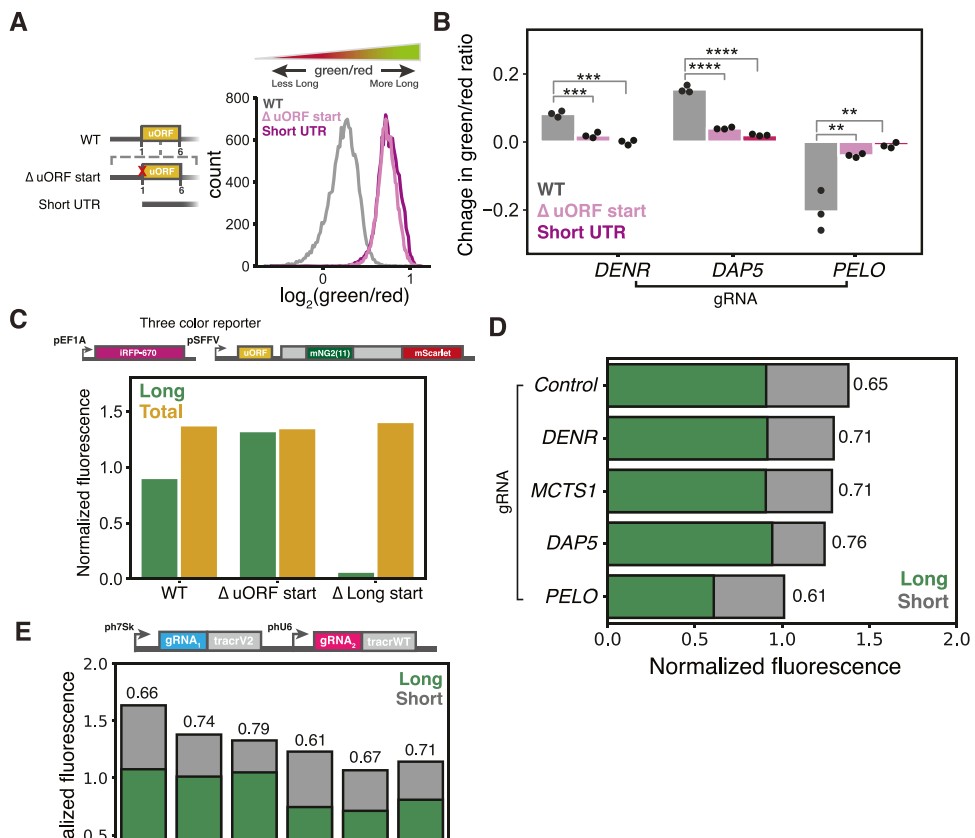

**Figure 4. Three-color reporter reveals that *PELO* knockdown decreases long isoform expression.**
**(A)** Flow cytometry measurements of the isoform ratio in reporters with no uORF start codon (Δ uORF) or a short unstructured 5′ UTR with no uORFs. **(B)** Flow cytometry measurements of the difference in median green/red ratio induced by CRISPRi knockdown, relative to nontargeting sgRNA (two-sided *t* test; **: < 0.01, ***: < 0.001, and ****: < 0.0001). **(C)** Three-color reporter with schematic and flow cytometry measurements normalized to iRFP670. **(D)** Long and short isoform expression with CRISPRi knockdown, calculated from iRFP670 normalization. The green/red ratio is indicated next to each knockdown. **(E)** As in (D), for single or double knockdown.

cytometry measurements and FACS enrichment across a collection of seven other sgRNAs (Fig 3B and Table S3). In contrast, sgRNAs against *PCBP2/hnRNPE2*, which uniformly represses translation of *CEBPA* (85), had no effect on the fluorescence ratio (Fig S3B and Table S3). We further confirmed that our top-scoring sgRNAs against *DENR* and *CNOT9* caused a more than 90% reduction in mRNA levels, validating the knockdown efficiency of our most enriched sgRNAs (Fig S3C).

Short isoform expression depends primarily on downstream reinitiation after uORF translation. In general, deacylated tRNAs must be released from 40S ribosomal subunits after termination to prepare them for a new round of initiation. The DENR/MCTS1 heterodimer promotes recycling of 40S ribosomal subunits by removing tRNAs after termination, thereby allowing the binding of new ternary complex and the resumption of scanning. By resetting 40S subunits, they could favor reinitiation after uORF translation (31, 32, 36, 70, 86), explaining the effects of *DENR* and *MCTS1* depletion on *CEBPA* start site selection.

The close connection between recycling and reinitiation also suggests how PELO—which participates in ribosome rescue and recycling after aberrant termination—could affect the choice between start sites. We first confirmed that the strongest sgRNA against *PELO* greatly reduced PELO protein levels (Fig 3C). We next

devised a complementation system to rescue *PELO* depletion by fusing a drug-controlled destabilization domain (DD) that is stabilized in the presence of the small molecule, Shield1 (87), to the PELO and HaloTag CDS (Fig S4A and B). We found that PELO overexpression—but not the overexpression of a HaloTag control—completely rescued the isoform ratio shift caused by *PELO* depletion (Fig 3D) and mitigated the substantial cell viability defect (Fig 3E). We recapitulated these results with a reporter containing the native Kozak sequence around the uORF start codon (Fig S4C and D), confirming that the effect did not depend on our modified uORF start site context. We also excluded the possibility that *PELO* depletion caused isoform-specific post-translational effects on protein stability by recapitulating its effects on a variant *CEBPA* reporter fused to a C-terminal, destabilizing PEST sequence that ensures the rapid turnover of both isoforms (Fig S4E). Instead, it appears that PELO plays an uncharacterized role in regulating translation start site selection on *CEBPA*.

### *PELO* knockdown decreases long isoform expression

As PELO has not previously been implicated in uORF-mediated regulation, we next asked whether its effect on the relative

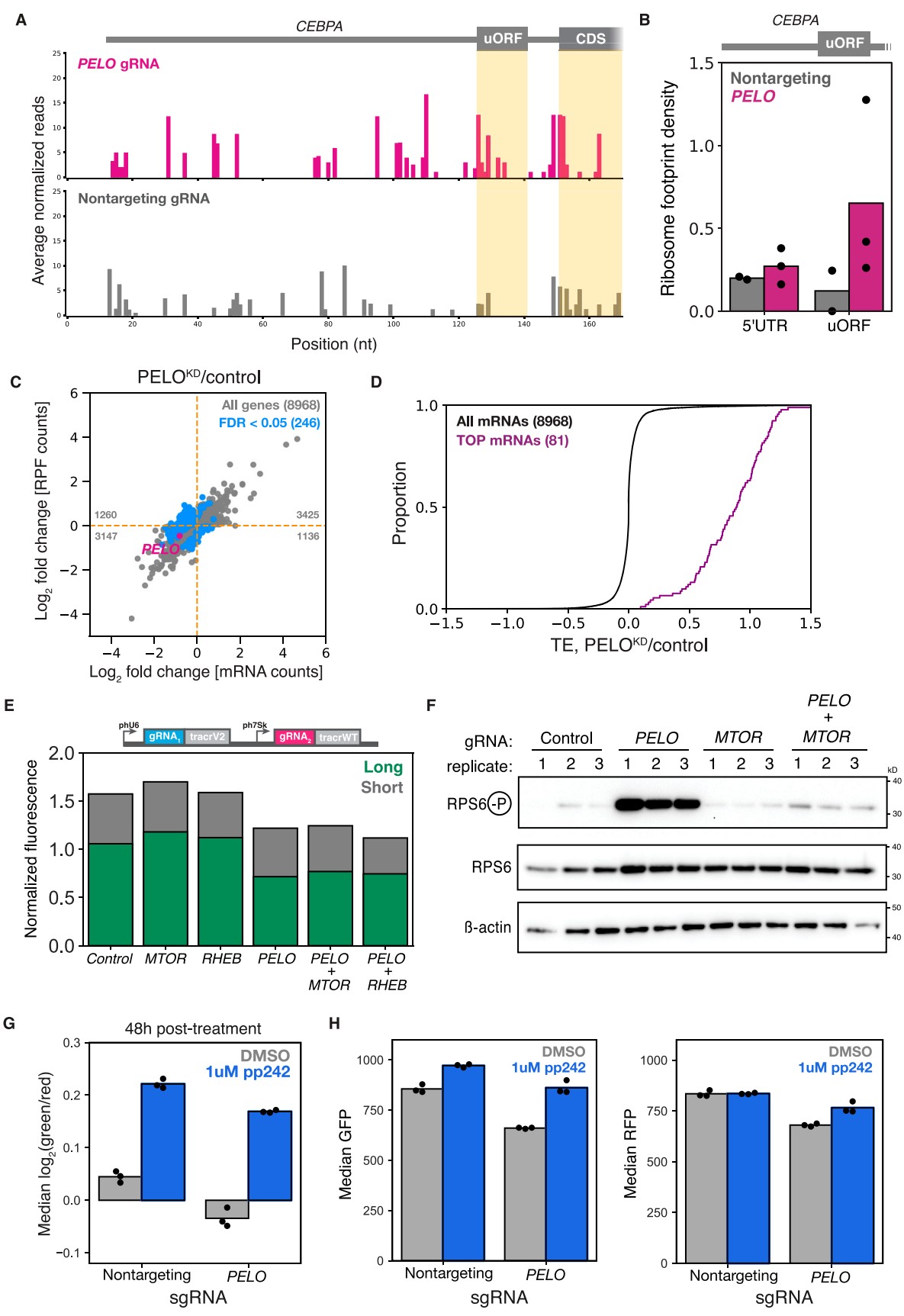

abundance of the two C/EBPα isoforms depended on reinitiation. We generated reporter variants with a mutation inactivating the uORF start codon (Fig 1D) or with a short unstructured 5′ UTR containing no uORFs, which should abolish reinitiation-dependent short isoform expression; indeed, we observed a higher green/red fluorescence ratio in these cell lines (Fig 4A). The effects of DENR and DAP5 knockdown were weakened or eliminated in these two reporters, which no longer support reinitiation, and the effects of PELO knockdown were likewise greatly attenuated in variants without uORFs (Fig 4B). Thus, the presence—and translation—of the CEBPA uORF is required for the change in isoform usage induced by PELO on our reporter.

Our two-color reporter provides a sensitive measure of changes in the isoform ratio, but does not distinguish whether PELO knockdown reduces long isoform expression or increases short isoform expression. To delineate between these possibilities, we expressed a third fluorescent protein that would serve as a normalizing control and allow us to quantify the absolute abundance of each CEBPA isoform. We constitutively expressed the infrared fluorescent protein iRFP670 (88, 89), which is spectrally distinct from mNeonGreen2 and mScarlet-I, enabling simultaneous quantification of all three fluorescent proteins by flow cytometry. We calibrated our fluorescence measurements using reporter variants that expressed only the long or short isoform because of substitutions in the uORF start codon or the long isoform start codons, respectively (Fig 1D). These normalized measurements revealed that our reporter produced 67% long isoform and 33% short isoform (Fig 4C).

We then characterized the effect of several sgRNA knockdowns on absolute isoform abundance. Consistent with their roles in promoting reinitiation at the short isoform start codon, depletion of DENR, MCTS1, or DAP5 led to statistically significant changes in short isoform expression with no marked change in long isoform levels. In PELO knockdown cells, we observed a ~30% decrease in long isoform expression, whereas short isoform abundance was largely unaffected, indicating that the shift in the green/red ratio is largely due to loss of long isoform (Figs 4D and S4F). Importantly, no sgRNAs significantly impacted iRFP670 (Fig S4F). These results affirm that DENR/MCTS1 and DAP5 mediate C/EBPα start site choice by driving reinitiation-dependent short isoform expression, whereas PELO plays an unexpected role in maintaining long isoform production.

We next investigated how these different factors interact by knocking down either DENR or DAP5 in combination with PELO using a dual-sgRNA expression vector (90) (Table S1). The effect of depleting DENR and PELO together was additive—knockdown of both genes reduced short isoform expression to the same extent as DENR knockdown alone, and long isoform expression to the

same degree as PELO single knockdown. Interestingly, depleting DAP5 and PELO together modestly weakened both the PELO-dependent loss of the long isoform and the DAP5-dependent loss of short isoform expression (Figs 4E and S4G). Nonetheless, the lack of strong epistasis between PELO and canonical reinitiation factors argues that PELO is not directly affecting reinitiation.

## PELO depletion increases CEBPA uORF translation and activates mTOR

To directly measure the translational effects of PELO depletion on our reporter, and across the transcriptome, we performed ribosome profiling (91, 92) in our reporter cell line transduced with either the top-scoring PELO sgRNA or a nontargeting control. Transcriptome-wide translation measurements correlated very well (r > 0.94) across biological triplicates and showed clear sgRNA-specific differences (Fig S5A and B). We saw the characteristic accumulation of footprints in the 3′ UTR in PELO knockdown that correspond to unrecycled vacant 80S ribosomes (Fig S5C), as has been previously reported in both yeast and humans (59, 60). We further observed a notable increase in ribosome occupancy in the 5′ UTR of our CEBPA reporter in PELO-depleted cells. These included footprints on the CEBPA uORF along with a surprising accumulation of footprints that mapped near the long isoform start codon (Figs 5A and B and S5D). Although these footprints could be derived from ribosomes initiating at this start codon, PELO depletion reduces long isoform production. Instead, these footprints could originate from vacant ribosomes that accumulate after uORF termination, analogous to the unrecycled ribosomes that build up on 3′ UTRs after main ORF termination in PELO knockdown. Indeed, quantification of ribosome footprint density revealed a slight increase across the long isoform exclusive region despite the overall decrease in translation of the reporter suggesting the presence of these unrecycled ribosomes (Fig S5E). These unrecycled ribosomes could occlude the long isoform start codon, providing a mechanistic explanation for reduced long isoform expression as a direct consequence of lower PELO rescue activity.

In addition to these CEBPA-specific effects, we observed a number of translational changes across the transcriptome when we knocked down PELO. We computed translation efficiency (TE) using matched mRNA abundance data and found that 226 genes displayed a significant (false discovery rate–adjusted $P ≤ 0.05$) TE difference in our PELO depletion (Fig 5C). Importantly, however, the expression of our mNeonGreen2$_{1–10}$, fragment was unchanged (Fig S5F). Among the genes with the strongest increase in TE, 37% (84 genes) were 5′ terminal oligopyrimidine motif–containing (TOP)

**Figure 5. PELO knockdown increases uORF translation and activates mTOR.**
**(A)** Ribosome occupancy profile of CEBPA 5′ UTR, normalized by CDS occupancy (n = 3). The average median count across replicates is shown. **(B)** Ribosome footprint density (footprint count/length) across the 5′ UTR and uORF of CEBPA. **(C)** Scatterplot of log$_2$ fold changes (PELO KD/control) in ribosome footprint and RNA-seq counts. Each point represents a single gene, and the total number of genes in each quadrant is indicated. Genes with statistically significant (false discovery rate < 0.05) change in translation efficiency are highlighted. **(D)** Cumulative distribution of TE changes of terminal oligopyrimidine (TOP)–containing mRNAs. **(E)** Long and short isoform expression after CRISPRi knockdown, as in Fig 4E. **(F)** Western blot of K562 cells after CRISPRi knockdown. Replicates represent independent transductions. **(G)** Median isoform ratio measurements after CRISPRi knockdown and treatment with DMSO or 1 μM PP242 (Sigma-Aldrich) for 48 h. **(H)** Flow cytometry measurements of median green and red fluorescence from (G), n = 3.

mRNAs (93), which primarily encode ribosomal and ribosome-associated proteins (Fig 5D). Translational up-regulation of 5′ terminal oligopyrimidine mRNAs is a hallmark of mTOR activation. Indeed, previous work in human fibroblasts and in mouse models also observed enhanced translation of mTOR-regulated transcripts in *PELO* knockouts (82, 94). mTOR activation has broad-ranging effects on protein synthesis (95, 96), and rapamycin-induced mTOR inhibition favors C/EBPα long isoform expression (18), raising the possibility that mTOR activation explains the reduced long isoform expression in *PELO* knockdown. We thus wanted to ask whether PELO affects C/EBPα translation above and beyond any mTOR-mediated changes.

Despite the known effect of mTOR activity on *CEBPA* translation, genes from this pathway did not stand out in our screen. We identified a modest but significant effect from a single one of the sgRNAs targeting the mTOR kinase, *MTOR*, itself and no significant effects from sgRNAs targeting mTOR regulators *RHEB* or *TSC1* (Fig S5G). Individual knockdown of *MTOR* led to a slight increase in the isoform ratio, much smaller than the change seen in *PELO* knockdown, consistent with the weak phenotype in our screen. Neither targeted knockdown of the mTORC1 activator *RHEB* nor the negative regulator *TSC1* shifted isoform levels (Figs 5E and S5G and H). We confirmed that CRISPRi depletion of *MTOR* substantially reduced phosphorylation of RPS6, an indirect translation-related target (Fig S5I). We also found that *PELO* knockdown led to a pronounced increase in RPS6 phosphorylation and thus mTOR activity. Simultaneous knockdown of both *PELO* and *MTOR* resulted in low RPS6 phosphorylation, similar to that seen in unperturbed cells and far below the levels in *PELO* depletion alone (Fig 5F). However, the *PELO* depletion effect on our reporter was unaffected by simultaneous knockdown of *MTOR* itself or these key regulators (Figs 5E and S5H). As *MTOR* knockdown greatly suppresses the effect of PELO on mTOR activity but does not modify its effects on the isoform ratio, PELO appears to play a specific mTOR-independent role in *CEBPA* translation.

Because CRISPRi produces partial loss-of-function phenotypes, we further tested how the *PELO* knockdown phenotype was affected under saturating chemical inhibition by PP242 (Fig S5J). PP242 treatment shifted the isoform ratio of our reporter by increasing long isoform expression at the expense of short isoform synthesis, with little change in total abundance (Figs 5G and H and S1C). The shift toward long isoform expression in PP242-treated cells persisted in *PELO* knockdown (Fig 5G). Conversely, *PELO* knockdown reduced the long isoform fraction in the context of PP242 treatment, relative to a nontargeting control in PP242, consistent with a direct mTOR-independent role of PELO in start site selection (Fig 5H). *PELO* knockdown also reduced total protein levels in PP242-treated cells, again suggesting that lack of PELO directly impairs long isoform synthesis—perhaps through physical obstruction by unrecycled ribosomes, an effect that cannot be suppressed by mTOR inhibition—and that PELO acts as an mTOR-independent regulator of translation start site choice on C/EBPα.

### PELO effects on *CEBPA* translation depend on both uORF and spacer length

Our ribosome profiling data suggested that in *PELO* knockdown, unrecycled ribosomes might accumulate on the *CEBPA* 5′ UTR and occlude the long isoform start codon. This occlusion would depend on the specific configuration of the uORF and the long isoform start codon in *CEBPA*, perhaps explaining why PELO has not been implicated in translational control of other uORF-regulated genes. The short seven-nucleotide separation between the end of the uORF and the long isoform start codon and the six-codon length of the uORF are conserved across *CEBPA* homologs, although their sequence varies (Fig 6A). We thus tested the effects of our knockdowns in variant reporters in which the distance between the uORF and long isoform start codon was extended.

Interestingly, lengthening this spacer to 14 or 17 nucleotides increased long isoform expression substantially, relative to the WT seven-nucleotide spacer, resulting in an increase in the green/red ratio (Figs 6B and S6A and Table S1). When we introduced sgRNAs targeting *PELO* into each of these reporter lines, we found that the effect of *PELO* knockdown was diminished on both the 14-nucleotide and 17-nucleotide intercistronic spacer reporters relative to the WT seven-nucleotide uORF reporter (Fig 6C).

We next interrogated the impact of changing the length of the uORF while preserving the distance between the uORF stop codon and the main CDS start codon (Table S1). Shortening the uORF to nine nucleotides increased short isoform production by 50% with little effect on long isoform levels, in line with the general observation that shorter uORFs confer higher reinitiation probability (97, 98). In contrast, extending the uORF to 30 nucleotides decreased short isoform production by 40%, further increasing the isoform ratio relative to the WT uORF reporter. We also observed a substantial increase in long isoform expression when we extended the uORF, suggesting that this change disrupted some inhibitory effect (Figs 6D and S6B). Overall, our data demonstrate that reinitiation efficiency on *CEBPA* is sensitive to uORF length akin to other well-characterized uORFs (36, 73). As in our variant spacer length reporters, we found that *PELO* knockdown had less of an effect on these variant uORF reporters. In contrast, *DENR* knockdown produced a similar effect on the isoform ratio across all reporter lines. The effect of *DAP5* knockdown was also abrogated in the long uORF reporter, recapitulating previous observations that showed that shorter uORFs were particularly DAP5-dependent (73) (Figs 6E and S6C). Notably, extending the intercistronic length diminished the effect of *PELO* knockdown to a greater extent than changing uORF length. Together, these results suggest that, indeed, the effects of PELO on *CEBPA* translation are especially reliant on the conserved positioning of the uORF and CDS.

Given the robust effect of *PELO* depletion on C/EBPα reporter transcripts, we next wanted to investigate whether it similarly influenced isoform levels of *CEBPB*, another member of the C/EBP protein family whose 5′ UTR also encodes a regulatory uORF that determines the expression of multiple protein isoforms. Like *CEBPA*, translation of the *CEBPB* uORF leads to reinitiation at a downstream start site that encodes a truncated isoform (18). The *CEBPB* uORF, however, is twice as long (36 nucleotides) as the *CEBPA* uORF and is only four nucleotides upstream of the primary start site. To model this regulation, we inserted the 5′ UTR of *CEBPB* upstream of our two-color reporter (Table S1) and found that this reporter, in contrast to the 5′ UTR of *CEBPA*, produced over 90% long isoform. Despite the low efficiency of reinitiation on this reporter, it was nevertheless sensitive to mutations in the uORF start codon.

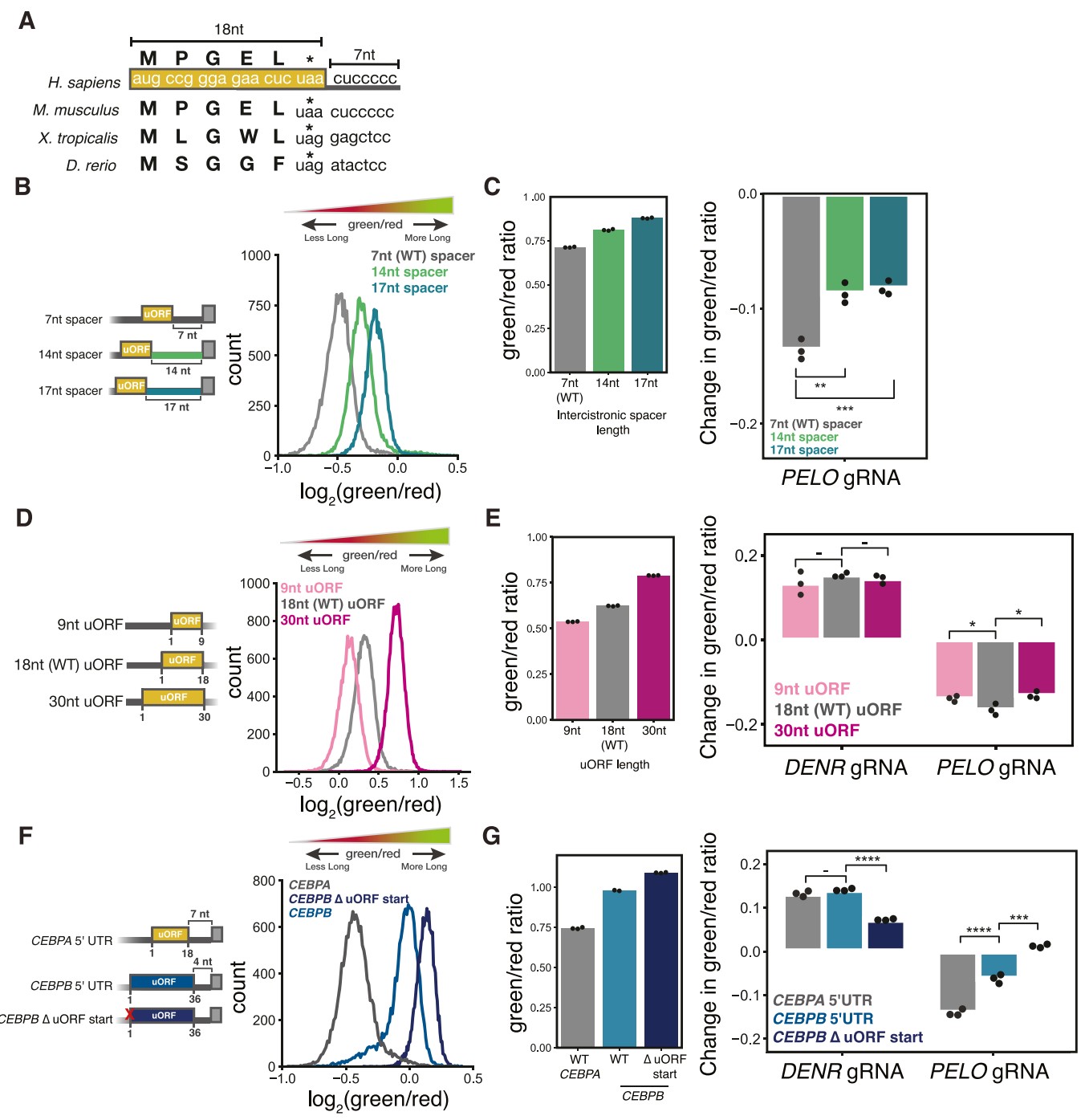

**Figure 6. Effect of PELO on *CEBPA* translation depends on uORF and intercistronic spacer length.**
**(A)** Schematic of *CEBPA* uORF. **(B)** Flow cytometry measurements of intercistronic spacer reporters. **(C)** Flow cytometry measurements of green/red ratio in control cells stably expressing each spacer reporter, n = 3 (left). Difference in median green/red ratio induced by CRISPRi knockdown, relative to nontargeting sgRNA in *CEBPA* spacer variant reporters (two-sided t test; *: < 0.05) (right). **(D)** As in (B), for uORF length variants. **(E)** As in (C), for uORF length variants (**: < 0.01 and ***: < 0.001). **(F)** As in (B), for *CEBPB*-derived reporter variants. **(G)** As in (C), for *CEBPB*-derived reporter variants (***: < 0.001 and ****: < 0.0001).

Disrupting uORF translation increased long isoform levels and critically abolished production of the small fraction of short isoform (Figs 6F and S6D).

We next tested the impact of our sgRNA-mediated depletions on these *CEBPB* reporters. Knocking down *DENR* increased the long isoform fraction on our *CEBPB* reporter to a similar extent as our *CEBPA* reporter. Depletion of *PELO* decreased the long isoform fraction, but this effect was twofold weaker than what we observed on our *CEBPA* reporter, perhaps owing to reduced reinitiation on this reporter. Both knockdown phenotypes were abrogated in a

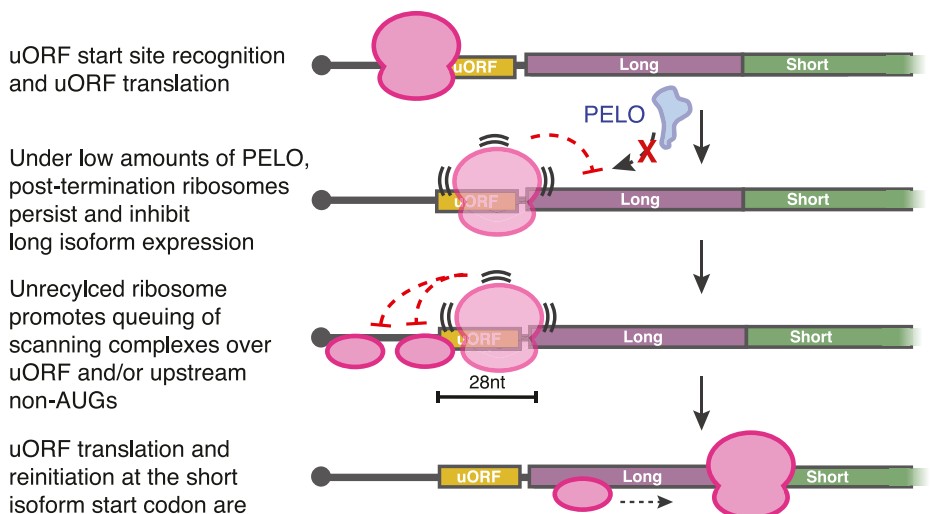

**Figure 7.  Model of PELO-mediated control of C/EBPα levels.**

uORF start site recognition and uORF translation

Under low amounts of PELO, post-termination ribosomes persist and inhibit long isoform expression

Unrecycled ribosome promotes queuing of scanning complexes over uORF and/or upstream non-AUGs

uORF translation and reinitiation at the short isoform start codon are preserved

variant *CEBPB* reporter with a mutation in the uORF start codon, further suggesting that similar to *CEBPA*, alternative start site usage depends on uORF translation (Figs 6G and S6E). Overall, these data suggest a role for PELO in mediating translation specifically of *CEBPA* and to a lesser extent *CEBPB*, perhaps owing to the unique sequence architecture of the *CEBPA* transcript leader.

The *CEBPA* 5′ UTR used in our reporter is sufficient to direct the regulated translation of distinct protein isoforms (18), and so *PELO* depletion is expected to shift the isoform ratio of endogenous C/EBPα. Immunoblotting measurements of endogenous protein lack the precision of fluorescent reporters, however, and K562 cells do not express *CEBPA* because they resemble later stages of myeloid differentiation. We thus knocked down *PELO* in monocytic THP1 cells expressing a dCas9-ZIM fusion (99) and provide evidence for reduced C/EBPα long isoform levels and unchanged, albeit low, levels of short isoform (Fig S7A). The magnitude of this shift was consistent with the effect seen in reporters using the WT *CEBPA* 5′ UTR (Fig S7B).

## Discussion

We survey the translation factors controlling the choice between two alternate start sites that produce opposing isoforms of the key hematopoietic transcription factor, C/EBPα. Known reinitiation factors DENR/MCTS1 and DAP5 all promote short isoform expression, consistent with the model that short isoform synthesis occurs through reinitiation after uORF translation. We also found that loss of ribosome rescue factor PELO, which has no described role in translation reinitiation or uORF-mediated regulation, reduces C/EBPα long isoform expression. Both PELO and DENR/MCTS1 have established molecular functions at the end of the translation cycle, and the impact of depleting these factors on *CEBPA* translation suggests that the fate of ribosomes after uORF translation controls start site choice for the downstream CDS. To

explain the surprising impact of ribosome rescue on C/EBPα translation, we propose that unrecycled ribosomes accumulate after uORF translation and block long isoform initiation, thereby influencing start site selection (Fig 7).

Our data argue that PELO impacts long isoform initiation specifically and does not affect reinitiation directly (Fig 4). Instead, in keeping with its function in other contexts, PELO likely removes unrecycled vacant ribosomes that would otherwise impede subsequent rounds of translation initiation at the long isoform start codon. Because of the conserved arrangement of the *CEBPA* uORF near this start codon, an 80S ribosome that is not removed from the mRNA by recycling after uORF translation would directly block the long isoform start codon. In addition, this post-termination ribosome could stall scanning preinitiation complexes near the uORF start codon, enhancing uORF translation. Accumulated post-termination ribosomes could also promote near-cognate initiation at non-AUG codons upstream of the uORF, which would also reduce long isoform expression specifically. On *CEBPA*, lowered PELO levels would therefore produce a self-reinforcing effect where a terminating or post-termination ribosome that is not removed after uORF translation would subsequently promote uORF initiation by the next scanning ribosome (Fig 7). Similar enhancements of upstream initiation are seen in other situations when scanning is blocked (45, 46, 47, 48). In support of this model, reporter variants that altered either the proximity of the uORF to the long isoform start codon or the length of the uORF decreased the requirement for PELO (Fig 6). Notably, however, PELO is recruited by unrecycled ribosomes themselves, and once recruited, it drives their removal from mRNAs (100), and so it is not possible to capture a direct association of PELO with substrate transcripts.

We further provide evidence that PELO's unexpected role in translation start site selection is mTOR-independent. We do see that loss of *PELO* activates mTOR signaling (Fig 5), as seen in other systems (82, 94). Impaired ribosome recycling after *PELO* depletion may reduce translational capacity (101), activating mTOR in a

compensatory response perhaps by the direct association of mTORC1 with ribosomes (102) or an imbalance between ribosome availability and protein biosynthesis capacity (101). Although such changes in mTOR activity can influence start site choice on *CEBPA* (18)—an effect we see as well (Fig 1)—loss of mTOR activity does not abolish the effect of PELO on our reporter (Fig 5).

We also show that *CEBPA* uORF translation is dependent on DENR/MCTS1 and DAP5 (Fig 4). The DENR/MCTS1 heterodimer recycles 40S subunits after termination (31, 32, 70, 86), and this activity is important for subsequent reinitiation on many transcripts, including *ATF4* (33, 36, 72). The need for DENR in post-termination 40S recycling varies according to the identity of the tRNA being evicted, and thus the penultimate codon (36); the *CEBPA* uORF ends with a Leu codon, which is expected to confer strong DENR dependence. The *CEBPA* 5′ UTR is highly structured as well, similar to other mRNAs that depend on DAP5 for efficient translation (73). Although the molecular details of DAP5 function are not clear, DAP5-dependent translation requires efficient termination and recycling, again connecting the end of uORF translation with downstream initiation.

C/EBPα is a critical hematopoietic transcription factor whose expression is regulated at multiple levels to ensure proper differentiation and control cell fate decisions. In the myeloid lineage, increasing long isoform expression over short drives granulocytic differentiation (103, 104), whereas the exclusive expression of the short isoform ratio can block this transition (105) and can promote leukemogenesis (51, 56, 106). Although the isoform ratio changes we observe after *PELO* depletion are more modest, they may affect these cell fate decisions if they are reinforced by coordinated regulation in related pathways or accompanied by sensitizing mutations. Ribosome rescue and recycling activity are themselves dynamically regulated during erythroid development, with initial up-regulation followed by a gradual decrease during differentiation (60). The expression of *PELO* is also decreased in AML (107), and our data suggest that this would favor the oncogenic short isoform. These trends underscore the broader physiological and pathological impact of PELO and the ribosome rescue pathway on the regulation of hematopoiesis.

# Materials and Methods

## Plasmid and two-color reporter construction

All plasmids and primers used are listed in Tables S1 and S2, respectively. All plasmids (with the exception of sgRNA expression vectors; see the section) were generated by the Gibson assembly (108) from amplicons made with primers indicated in Table S2.

In brief, to construct the *CEBPA* two-color reporter, the mNG2$_{11}$ sequence was subcloned from a pCMV-mNG2$_{11}$-H2B plasmid kindly gifted by Siyu Feng (63) and assembled into codon position 45 in the N-terminus of *CEBPA*. All Met codons in mNG2$_{11}$ were removed to avoid generating new in-frame start sites and substituted with either Val or Ile to preserve hydrophobicity. Similarly, an in-frame start codon at position 14 in *CEBPA* was mutated to ACA (Δ2) to ensure that only the long and short start sites were used.

Furthermore, the basic DNA-binding domain was deleted (ΔDBD) to suppress cell proliferation arrest caused by the ectopic expression of *CEBPA* (18, 109) and the Kozak sequence around the uORF start codon was optimized (gccgccATGg, as in Calkhoven et al (18)) to increase the dynamic range of our reporter readout. Human *CEBPA* cDNA was amplified from genomic DNA. Amplicons corresponding to the SFFV promoter, *CEBPA* 5′ UTR, CEBPA-mNG2$_{11}$, and CEBPA-mScarlet-1XFlag were then introduced into the SbfI and KpnI cut sites of the pNTI620 vector.

## Cell culture

Human K562 cells were grown in RPMI 1640 with L-glutamine (Thermo Fisher Scientific) supplemented with 10% FBS, 1% sodium pyruvate, 100 units/ml penicillin, and 100 mg/ml streptomycin. Human HEK293T Lenti-X cells were grown in DMEM + GlutaMAX (Thermo Fisher Scientific) supplemented with 10% FBS, 1% Hepes, 100 units/ml penicillin, and 100 mg/ml streptomycin. K562 cells were maintained at a cell density of $0.5 \times 10^6$/ml. All cell lines were obtained from the UC Berkeley Cell Culture Facility and grown at 37°C and 5% $CO_2$.

## Generation of stable two-color reporter cell lines

The mNG2$_{1-10}$ fragment was a gift from Siyu Feng (63), was transfected using TransIT-LTI Transfection Reagent (Mirus), and packaged with pNTI673 and pNTI674 (Table S1) in a HEK293T Lenti-X cell line to generate lentiviral particles. mNG2$_{1-10}$ was then stably integrated into a polyclonal dCas9-KRAB CRISPRi K562 cell line (66) by lentiviral transduction. Dual-color *CEBPA* reporter constructs (WT, ΔuORF start, Δlong start, short UTR; Table S1) were then stably integrated into this cell line by Cas9-mediated integration into the *AAVS1* locus by simultaneous nucleofection of plasmids containing either targeting sgRNAs (AAVS1-T2 and AAVS1-T2; Table S3) or spCas9 (110) and selected using 1 μg/ml puromycin (InvivoGen) to generate stable integrants. All lines were subsequently monoclonally isolated.

## Generation of stable three-color reporter cell lines

An iRFP670 construct was stably integrated into the CRISPRi (mNG2$_{1-10}$) cell line by Sleeping Beauty transposition (111) (Sleeping Beauty expression vector; Table S1), then selected using 800 μg/ml G418 (InvivoGen) followed by monoclonal isolation. Generation of three-color reporter lines was achieved by stable integration of dual-color reporter constructs into the *AAVS1* locus in this cell line (as above).

## Flow cytometry

Cells were harvested for flow cytometry analysis by centrifugation (500*g* for 5 min at RT) followed by resuspension in PBS supplemented by 1% FBS and 1 mM Hepes. All analyses were done on LSR Fortessa Analyzer (BD Biosciences). Cells were initially gated on forward scatter (FSC) and side scatter (SSC) (Fig S1A), and positive events were determined by a threshold based on negative (no fluorescence) and single-color control cells. Green (mNG2), red

(mScarlet), and iRFP (iRFP670) fluorescence was detected on the FITC (530/30 nM), PE/Texas Red (610/20 nM), and APC-Cy7 (780/60 nM) channels, respectively. The green/red fluorescence ratio was calculated by taking the ratio of absolute mNG2 fluorescent values to the absolute mScarlet fluorescent values per cell. To determine the normalized fluorescent values in the three-color reporter assays, total levels of long and short isoforms were obtained by measuring green and red fluorescence in variant reporter lines expressing only either the long or short isoforms because of mutations in either the uORF start codon to abolish short isoform expression or the long isoform start codon. These fluorescent values were then divided by iRFP levels per cell to obtain normalized green and red fluorescence levels.

### FACS-based CRISPRi screen

CRISPRi sublibrary screens were performed using four compact BFP-tagged CRISPRi sublibraries containing five sgRNAs per TSS (Cat#83971-3 and #83975; Addgene) expressed in the pCRISPRi-v2 expression vector (Cat#84832; Addgene). Plasmid sublibraries were separately packaged in HEK293T Lenti-X cells and transduced into the CRISPRi dual-color reporter line at an MOI < 1 where the percentage of transduced cells by BFP expression after 2 d post-transduction was 20–30%. At 2 d post-transduction, we performed FACS using an Aria Fusion (BD Biosciences) cell sorter to select for cells expressing BFP. Cells with the highest (~20%) BFP expression were collected and recovered in RPMI 1640 for 6 d post-FACS. ~10 million cells were collected per sublibrary, maintaining an average sgRNA coverage of at least 500 cells per sgRNA.

At 6 d post-BFP selection, cells were again sorted using a FACS Aria Fusion cell sorter based on the ratio of green to red fluorescence from our CEBPA dual-color reporter line. ~40 million cells per sublibrary transduction were sorted into four distinct green/red bins (~20–25% of cells in each bin), with each bin containing ~8–10 million cells to ensure an average sgRNA/cell coverage of at least 500. Genomic DNA was immediately harvested from these cells using the DNeasy Blood and Tissue kit (69504; QIAGEN), and sgRNA fragments were isolated by SbfI (New England Biolabs) restriction digestion and AMPure bead size selection, then amplified by PCR for deep sequencing as described in reference (67). The sgRNAs were sequenced on an Illumina HiSeq 4000 using custom primers. Sequencing data are deposited with NCBI GEO, accession GSE226435.

### CRISPRi screen processing and data analysis

Sequencing reads were trimmed to remove adapter sequences using Cutadapt (112), and trimmed sgRNAs were counted using MAGeCK (113, 114). Raw sgRNA counts were then used as input to DESeq2 (115) to calculate enrichment scores (isoform shift scores) in which each fluorescent bin (FR, NR, NG, FG) was represented as a numeric covariate in the linear model such that

$$bins = (-1, 0, 0, 1)$$

This assumes that sgRNA counts in the FR and FG bins (at the extremes of the fluorescence ratio distribution) have a constant multiplicative change with respect to the middle (NR and NG) bins. Raw read counts per bin and enrichment scores for all sgRNAs across all sublibraries are provided as Supplementary Information.

Gene ontology analysis was performed using PANTHER (116, 117, 118) with background lists representing the genes targeted by each hCRISPRa-v2 sublibrary.

### Individual validation of sgRNA-mediated phenotypes

Individual sgRNA expression vectors were cloned by first annealing complementary synthetic oligonucleotide sequences containing each sgRNA protospacer (Table S3) flanked by BstXI (New England Biolabs) and BlpI (New England Biolabs) restriction sites (Integrated DNA Technologies). Each double-stranded annealed pair was then ligated into a BstXI/BlpI-digested pCRISPRi-v2 expression vector containing a BFP cassette. Each sgRNA expression vector was then packaged into lentiviruses in HEK293T Lenti-X cells and was individually transduced into either two- or three-color CEBPA CRISPRi reporter lines at an MOI < 1, resulting in ~20–30% infected cells by BFP expression. Cells were sorted on BFP 2 d post-transduction to select for sgRNA expression and allowed to recover in RPMI 1640 for 6 d. Reporter expression was then measured by flow cytometry.

### Western blots

Cells were collected by centrifugation (500$g$ for 5 min at RT), washed once with PBS, centrifuged again, and lysed with buffer (140 mM KCl, 10 mM Hepes, 5 mM MgCl$_2$, 1% Triton X-100, 1 mM TCEP, 2 U/μl Turbo DNase) on ice for 30 min. Lysates were then clarified by centrifugation (20,000$g$ for 10 min at 4°C). Protein lysates were separated on Bolt 4–8% Bis–Tris gels (Thermo Fisher Scientific), then transferred onto nitrocellulose membranes. Membranes were blocked with 5% milk in TBST (0.05% Tween-20) for 1 h at RT. Primary antibodies were incubated overnight at 4°C, and secondary antibodies, for 1 h at RT. C/EBPα protein was probed using a primary C/EBPα antibody (1:1,000, #2295; Cell Signaling Technology), V5 epitope tags were probed by a primary V5-tag antibody (1:2,000, #13202; Cell Signaling Technology), and β-actin loading controls were probed by a primary β-actin conjugated to HRP (1:2,000, #12620; Cell Signaling Technology). An HRP-conjugated anti-rabbit IgG (1:2,000, #7074; Cell Signaling Technology) was used as a secondary antibody against all primary antibodies. All blots were developed with SuperSignal West Dura Extended Duration Substrate (Thermo Fisher Scientific) and were visualized by a FluorChem R imaging system (ProteinSimple).

### PELO re-expression rescue assays

The HaloTag and PELO CDS was fused to the FKBP12 destabilization domain (DD) to generate (N-terminal) DD-HaloTag and PELO-DD (C-terminal) constructs. DD-containing constructs are stabilized by binding to the small molecule, Shield1 (87), such that in the absence of Shield1, the resultant proteins are rapidly degraded. These constructs were stably integrated into K562 CRISPRi dual-color reporter lines by Sleeping Beauty transposition (111) (Sleeping Beauty expression vector; Table S1). Polyclonal cells were selected

using 200 µg/ml hygromycin (InvivoGen). Single sgRNAs were then packaged and transduced (as described above) into these cell lines and sorted for BFP-tagged sgRNA expression. Cells were then treated with 1 µM Shield1 (Cat#632189; Takara Bio) 5 d post-BFP sorting, then harvested for flow cytometry and Western blot analysis 3 d post-Shield1 treatment. Because of the incomplete destabilization of the PELO-DD construct (because of the necessity of placing the DD-tag at the C-terminus), we were unable to use a noninduced condition as a point of comparison (Fig S4A).

### Ribosome profiling and RNA sequencing

For ribosome profiling and matched RNA sequencing, K562 CRISPRi dual-color reporter cells were first transduced with either a non-targeting sgRNA or the top-scoring *PELO* sgRNA (see individual sgRNA knockdown validation) in triplicate and were grown in T150 flasks (Corning) for 6 d post-BFP selection. Cells (5.0 × 10^6 non-targeting sgRNA and 2.5 × 10^6 *PELO* sgRNA per replicate) were harvested as previously described (92) without the addition of cycloheximide, and a sample of the lysate was taken for RNA sequencing. We used *E. coli* RNase I (10 U/µl; Epicentre) to generate *CEBPA* ribosome density profiles as per reference (92), whereas genome-wide translation measurements were derived from P1 nuclease (100 U/µl; New England Biolabs)–treated ribosome profiling libraries as described in reference (119). Briefly, nuclease-treated lysates were placed in 1M D-sucrose followed by ultracentrifugation for 1 h at 100,000 RPM at 4°C in a TLA-110 rotor to collect ribosomes. Footprints were resolved on a denaturing 15% Urea–PAGE, and RNase I footprints were size-selected from 17 to 34 nt, then dephosphorylated. Similarly, P1-digested footprints were size-selected from 30 to 40 nt. Footprint fragments were then ligated to preadenylated DNA linkers with T4 Rnl2(tr) K227Q (200 U/µl; New England Biolabs). Next, ligations were reverse-transcribed using ProtoScript II (200 U/µl) and circularized with CircLigase I (100 U; Epicentre) for 3 h at 60°C. Sequencing libraries were prepared from circularized cDNA templates as described in reference (92).

For matched RNA sequencing, cells were harvested by phenol–chloroform extraction and processed according to NEB Ultra II Directional RNA Sample Prep Kit (#E7760S; New England Biolabs).

The ribosome profiling and total RNA-sequencing samples were sequenced on an Illumina NovaSeq instrument. Ribosome profiling data are deposited at NCBI GEO, accession GSE226436.

### Ribosome profiling sequencing analysis

Ribosome profiling reads were first trimmed using Cutadapt and aligned to ribosomal RNA (rRNA) and transfer RNA (tRNA) references using Bowtie2 (120). The remaining reads were subsequently aligned to the transcriptome using STAR (121). Transcriptome-based alignments were then filtered to exclude the first 15 and last 5 codons because of the accumulation of initiating and terminating ribosomes. Finally, footprint abundance was quantified from these alignments by RSEM (122). Differential expression and translation efficiency analyses were conducted using DESeq2 (115).

## Data Availability

Raw and processed high-throughput sequencing data from the CRISPRi sublibrary screens and ribosome profiling have been deposited in GEO and are publicly available under accession numbers GSE226435 and GSE226436, respectively.

## Supplementary Information

## Acknowledgements

We thank J Wren Kim, Joe Lobel, Paige Diamond, and Liana Lareau, and members of the Ingolia and Lareau laboratories for invaluable scientific discussions. mNeonGreen2 plasmids were kind gifts from Siyu Feng. We also thank Jonathan Weissman's laboratory for the K562 CRISPRi cell line. Hector Nolla, Alma Nuguid Valeros, and Kartoosh Heydari provided indispensable support at the Flow Cytometry Facility at UC Berkeley. We also thank the Vincent J Coates Genomics Sequencing Laboratory at UC Berkeley, the UC Berkeley DNA Sequencing Facility, and the UC Berkeley Cell Culture Facility. This work was supported by the National Institutes of Health (www.nih.gov) grants DP2 CA195768 and R01 GM130996 (NT Ingolia) and Shared Instrumentation Grant S10 OD018174. The funders had no role in study design, data collection and analysis, decision to publish, or preparation of the article.

### Author Contributions

SG Fernandez: conceptualization, data curation, formal analysis, validation, investigation, visualization, methodology, and writing—original draft, review, and editing.
L Ferguson: software.
NT Ingolia: conceptualization, supervision, funding acquisition, methodology, project administration, and writing—original draft, review, and editing.

### Conflict of Interest Statement

NT Ingolia declares equity in Tevard Biosciences and Velia Therapeutics.

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
