## [Reviewer comments · Life Science Alliance]

Life Science Alliance

Ribosome rescue factor PELOTA modulates translation start site choice for C/EBP α protein isoforms

Samantha Fernandez, Lucas Ferguson, and Nicholas Ingolia

DOI: <https://doi.org/10.26508/lsa.202302501>

Corresponding author(s): Nicholas Ingolia, University of California, Berkeley

Review Timeline:

Submission Date:	2023-11-28
Editorial Decision:	2024-01-02
Revision Received:	2024-03-25
Editorial Decision:	2024-03-27
Revision Received:	2024-04-15
Accepted:	2024-04-16

Transaction Report:

January 2, 2024

Re: Life Science Alliance manuscript #LSA-2023-02501-T

Prof. Nicholas Ingolia
University of California, Berkeley
Molecular and Cell Biology
1 Barker Hall # 3202
Berkeley, CA 94720

Dear Dr. Ingolia,

Thank you for submitting your manuscript entitled "Ribosome rescue factor PELOTA modulates translation start site choice and protein isoforms of C/EBP α " to Life Science Alliance. The manuscript was assessed by expert reviewers, whose comments are appended to this letter. We invite you to submit a revised manuscript addressing the Reviewer comments.

Thank you for this interesting contribution to Life Science Alliance. We are looking forward to receiving your revised manuscript.

Sincerely,

B. MANUSCRIPT ORGANIZATION AND FORMATTING:

Reviewer #1 (Comments to the Authors (Required)):

In this manuscript, the group of Nicholas Ingolia reports the discovery that PELO is a new player in translation reinitiation and ribosome recycling. They find PELO is key to regulating the synthesis of two alternative polypeptides during the translation of C/EBPalpha mRNA. The two isoforms derive from two different start sites. Overall, the paper moves ahead in a field hardly explored in mRNA translation initiation, namely the reinitiation process of translation and what factors are involved in it. For all conclusions, the authors show strongly supportive data. The paper is well written and the experiments are well performed. Before publication, some minor points should be addressed: 1) Authors should explain (introduction) what is the difference, in terms of mechanism, between reinitiation and ribosome recycling. 2) In the Abstract, they should correct the definition of mTOR. "mTOR" stands for mammalian target of rapamycin, not "mechanistic target of rapamycin. 3) In Figure 1 or Figure 6A, the authors should include the complete Kozak sequence of the two alternative ORF, from the -3 to the +4 nucleotides surrounding both AUG start codons. 4) Is there evidence that misregulation of the translation of the two alternative ORFs of C/EBPalpha has actual consequences in hematopoiesis and leukemogenesis? It would be worth performing a survey in the reported proteomes of leukemia to see if there is evidence of unregulated synthesis of the two alternative C/EBPalpha polypeptides.

Reviewer #2 (Comments to the Authors (Required)):

CEBPA is a transcription factor involved in cell differentiation of multiple tissue lineages. Its expression is regulated at different levels, including during protein synthesis. uORF-dependent control of start codon choice results in the production of alternative protein isoforms from a single CEBPA mRNA. The molecular players in the translational control of CEBPA mRNA remain unknown. As disruption of the CEBPA protein isoform ratio is associated with human cancers, elucidation of the underlying molecular mechanisms may uncover new therapeutic strategies.

In this study, the authors performed a CRISPRi screen using a dual fluorescent reporter mimicking CEBPA mRNA translational control. With this experiment, Fernandez et al. found multiple hits that caused shifted CEBPA isoform ratio. Among the factors producing the strongest and most significant changes in isoform ratio were translation factors known to regulate reinitiation (DENR, eIF3H and eIF4G2) and the ribosome rescue factor Pelota (PELO). The unexpected association of a ribosome recycling factor with uORF-mediated regulation of protein synthesis prompted the authors to understand the role of PELO in start codon choice on the CEBPA mRNA. Based on their results, the authors add a new layer in the control of CEBPA isoforms expression in cells. Following uORF translation, PELO is required to recycle and remove post-termination ribosomes obstructing the start codon required for the synthesis of the long CEBPA isoform. Thus, in conditions of decreased PELO activity, the ratio of CEBPA isoforms is altered.

The experiments presented in the manuscript are elegant, solid and well performed. Attesting the quality of their experiments, several of the results support previous studies addressing the translational regulation of CEBPA mRNA and the role of PELO, DENR and eIF4G2 in translation. There are however, several concerns that could be addressed to further support the proposed model on the regulation of CEBPA mRNA translation by PELO.

1. The authors propose that PELO enhances long CEBPA isoform expression by recycling and removing ribosomes that were engaged in uORF translation and that block start codon recognition. However, none of the experiments showed that PELO associates with the 5' UTR of PELO mRNA, i.e., with the post-termination ribosomes following uORF translation. An experiment addressing the PELO binding to the 5' UTR of the CEBPA reporter (with and without uORF) might show the specific association of PELO with these post-termination ribosomes.
2. Wouldn't it be reasonable to assume that in PELO KD conditions, the excess of unrecycled ribosomes in the 5' UTR would increase the chances of reinitiation and therefore the synthesis of the short CEBPA isoform? However, PELO KD seems only to affect the expression of the long CEBPA isoform. Based on the ribosome profiling data, DENR and eIF4G2 levels are not significantly altered upon PELO knockdown. Can the authors comment on this observation?
3. Figure 5 and Supplementary Figure 5 - the ribosome footprint density along the CEBPA reporter mRNA in PELO gRNA cells seems to be increased not only in the vicinity of the uORF and the start codon of the long isoform, but across the 5' UTR and CDS (Supplementary Figure 5D). Is this correct? It is a bit difficult to assess the difference between control and PELO gRNA cells in Supplementary Fig. 5D (too small). Can the authors please provide some other type of figure/quantification showing the differences in ribosome footprints along the CEBPA mRNA in control and PELO gRNA conditions?
4. If the hypothesis proposed by the authors on the function of PELO in CEBPA mRNA start codon choice is correct, the increased ribosome footprints around the long isoform start codon in PELO gRNA cells should disappear or no longer

concentrate at this position in the CEBPA reporter mRNA that lacks the uORF start site, since in this condition the expression of the long isoform increases and PELO is no longer required. Can the authors show this effect on the delta uORF start reporter mRNA? Alternatively, in a previous study (Mills et al. Cell Reports 2016) the authors addressed the presence of unrecycled ribosomes in the 3' UTR by treating cell lysates with high salt concentrations to release vacant ribosomes. A similar treatment with the CEBPA mRNA from PELO gRNA treated cells would show that the footprints observed at the long isoform start site correspond to post-termination ribosomes.

Minor points:

- A detailed description of the calculation used to determine the relative abundance of the two translational isoforms by fluorescence ratio would benefit the reader and interpretation of the results. Are fluorescence absolute values used, or median values? This information could be provided in the material and methods section.
- In the knocked down experiments performed in the manuscript the authors have only showed that CRISPRi gRNAs reduced PELO protein levels. Other factors were also downregulated (e.g.: DENR, eIF4G2...) but the efficiency of the transcriptional inhibition on protein levels is not shown in any of the experiments. In addition, complementation experiments were only performed for PELO. Can the authors also rescue the effects on start codon choice with DENR and eIF4G2?
- In page 8, the authors refer to other studies where PELO KO/KD cells/tissues have been used in conjunction with ribosome profiling. Have the authors checked the footprint densities on endogenous CEBPA and CEBPB mRNAs (in case the mRNAs were expressed in the profiled tissues), and is the footprint distribution similar to the one described in the manuscript?
- The authors have used a split, self-complementing green fluorescent protein mNeonGreen2 system to score green fluorescence. Can changes in mNG2 1-10 expression levels following CRISPRi indirectly affect the green fluorescence measured in the experiments? Since RNA-seq was performed, can the authors show that in the positive hits associated with decreased green fluorescence, mNG2 1-10 mRNA levels remain unaltered?
- The authors should refer to the supplementary tables in the text of the different sections of the manuscript.
- The authors refer in page 11 that expression of PELO is decreased in AML. Can they corroborate that AML cell lines have higher levels of the short CEBPA isoform using western blotting analysis?
- Figure 2 - how would the mTOR inhibitor PP442 and the delta uAUG results score in these bins?
- Fig. 2C and D - data could be better understood if the different bins (FR, NR, NG, FG) would also be indicated in the graph (dashed lines or shadows). In addition, the supplementary tables with the scores for the different gRNAs could also benefit if bin information was also present for each gene with a significant effect on isoform ratio shift (or indicate genes that enhance long or short isoform expression).
- Fig. 2E and F - have the authors also analysed the GO terms enriched in the sgRNAs that produced a significant shift towards short isoform?
- A western blotting analysis of CEBPA isoforms in PELOTA KD conditions (Fig. 3) would allow the reader to understand the shift in isoform expression. How much of long isoform is still produced?
- Fig. 4D and E would benefit the interpretation of the results if the % of long and short isoforms in total CEBPA protein was also displayed in the graphs. The total normalized fluorescence levels change between conditions making it difficult for the reader to compare all the conditions.
- Fig. 5B - The information of the numbers of genes present in each quadrant could be presented.
- Fig. 6B, D and F - quantification (bar graph) of the effect of uORF size, spacer and 5'UTR in control cells would help the reader to understand the effect of each parameter in the green/red ratio.
- Sup Fig.1 - x axis scale is missing. Is overall translation also inhibited? Overall red fluorescence decreased or not? What is the ratio of green in the total red? Or, is only the ratio of green to red that it is altered?
- Sup Fig. 4B - lane 9 of WB is more similar to lanes 10-12 than to lanes 7-8. Is the blot mislabelled?
- Sup Fig. 4 - a short explanation of the shld 1 effect on PELO would benefit the reader (for instance in the figure legend or material and methods).
- Protein size ladder missing in many of the WBs (kDa)
- Sup Fig. 7 - This result would be easier to analyse if the protein bands were better separated. It looks like two long isoform bands are present and that only of these is reduced following PELO knock down. Can the authors repeat the blot with a gel that runs longer or of different acrylamide percentage to better discriminate the different CEBPA isoforms expressed in these cells? In addition, a scheme describing the different start sites originating the multiple CEBPA isoforms in THP1 cells would be helpful.

Reviewer #3 (Comments to the Authors (Required)):

This is an interesting paper that uses a very creative forward selection involving a 2-color reporter and FACS analysis to screen CRISPRi libraries in mammalian cells to identify trans-acting factors affecting uORF-mediated translational control of the C/EBPa gene. They identify DENR and DAP5 and provide evidence from reporter experiments that they act by enhancing reinitiation following translation of the uORF, which leads to increased initiation further downstream from the next AUG codon initiating the long C/EBPa isoform at the subsequent AUG codon for the short isoform. These results are consistent with previous findings on these two factors in promoting reinitiation after uORF translation; although in the case of DENR, the stimulatory effect has been found to be highly dependent on the last, penultimate codon of the uORF, which the authors did not address. Importantly, they also implicate for the first time the ribosome dissociation factor PELO in controlling the effects of a regulatory uORF in blocking initiation at a downstream CDS start codon. Their reporter data indicate that depleting PELO reduces expression of the C/EBPa long isoform without impacting reinitiation at the downstream short isoform AUG codon, suggesting (at least to this reviewer) that

leaky scanning through the uORF AUG codon, which is necessary to initiate at the second AUG for the long isoform, is impaired. Ribosome profiling data of the native gene on PELO depletion shows increased RPF occupancies in the uORF, which would fit with increased uORF translation and decreased leaky scanning to the long isoform AUG; however, it also gives the unexpected result of increased 80S ribosome occupancies at the long isoform AUG codon, at odds with the conclusion from reporter data that initiation at the long isoform AUG is reduced. They attempt to explain this by proposing that the increased RPFs at the long uORF AUG codon represent vacant 80Ss that escaped recycling by PELO after terminating at the uORF stop codon and that they interfere with initiation by the 40S preinitiation complexes (PICs) that leaky-scan through the uORF AUG codon. They attempt to support this mechanism by showing that lengthening or shortening the uORF, or increasing the separation between the uORF stop codon and long isoform start codon diminish the impact of depleting PELO on long isoform initiation; however, it's not clear to this reviewer how these findings would support their model. They present western blot evidence that PELO depletion reduces expression of the native long isoform of C/EBPa mRNA, and also reporter data indicating that PELO functions in the same manner for the similarly regulated C/EBPb mRNA. Consistent with previous data, depletion of PELO activates mTOR, which was shown previously to decrease the ratio of long:short C/EBPb isoforms; however, they show that PELO depletion reduces long isoform translation independently of its activation of mTOR by showing that the reduction in long isoform translation conferred by PELO remains evident after depletion of mTOR or RHEB.

The results implicating PELO for the first time in controlling translational control by the uORFs in the C/EBPa and b mRNAs in a manner that specifically reduces translation of the long isoforms are convincing and represent an important accomplishment.

-However, I am not convinced of the proposed mechanism. The retention of vacant 80S ribosomes at the uORF stop codon that would physically obscure the AUG codon of the long isoform is not supported by the profiling data, which show no 80S peak at the uORF stop codon. Instead, the elevated 80S peak is located at the AUG of the long isoform. It's possible, as they allude to, that these are vacant 80S complexes that have migrated from the uORF stop codon to the AUG of the long isoform, but this does not seem to be their model based on its description in the DISCUSSION where they indicate that an 80S ribosome that is not released from the uORF stop codon after uORF translation would directly block access to the long isoform start codon because of the close spacing between the stop and start codons. The other problem is that if a vacant 80S is blocking initiation at the long isoform AUG codon by any 40S PICs that leaky-scanned the uORF then it should also block progression of these scanning PICs to the next AUG for the short isoform and thereby reduce reinitiation at the latter start site.

-I also don't understand how their model is supported by the observation that the impact of PELO depletion is reduced by changing the uORF length, as these changes should not alter the retention of unrecycled 80Ss at the uORF stop codon.

-And while I follow that increasing the spacing between the uORF stop codon and AUG for the long isoform from 7 nt to 14nt or 17nt would allow a PIC to form at the long isoform AUG even if an unrecycled 80S is still positioned at the uORF start codon, I don't see how the scanning PIC can make it through the unrecycled 80S to benefit from the expanded separation.

-The authors might consider instead that PELO depletion reduces leaky scanning of the uORF AUG codon to reduce the number of ribosomes able to scan past the uORF to the long isoform AUG, but in a manner that does not lead to increased reinitiation at the short isoform AUG. Increased initiation at the uORF AUG is supported by the profiling data, which also suggests increased initiation at near cognate start sites upstream of the uORF. Perhaps, retention of an unrecycled 80S at the stop codon on PELO depletion leads to a translation pause in the uORF that slows down scanning PICs approaching the uORF. This leads them to initiate aberrantly at near-cognates upstream of the uORF, and also to increase their probability of recognizing the uORF AUG to thereby reduce leaky scanning to the long isoform AUG. (The authors have already mentioned this type of queuing in the uORF that would reinforce initiation there.) If one proposes that the net outcome of slower elongation and increased initiation in the uORF yields essentially the same number of 80S ribosomes that complete uORF translation and go on to reinitiate at the short isoform AUG, this would lead to no reduction in short isoform initiation and explain the observed specific reduction in long isoform translation. If the unrecycled 80S tends to migrate away from the uORF stop codon and stall at the long isoform AUG without completing initiation, and then moves on or falls off the mRNA, this would explain their profiling data.

-Two other issues with the profiling data are that we are not being shown replicates, and we are not being shown the region of the main CDS up to and beyond the AUG for the short isoform to evaluate whether the expected reduction in RPF density between the AUGs for the long and short isoforms is actually evident on PELO depletion. If it is not, then the utility of the profiling data for revealing the mechanism is questionable.

-Fig. 1D: it is expected that deletion of the uORF start will increase long isoform production, but this blot seems to show reduced long isoform instead, unless this is a loading problem. It's shown later in Fig. S6D, that this predicted effect is observed on removing the uORF for C/EBPb for the normalized reporter. I suggest they include a similar figure for C/EBPa and note in the text that the expected result will be shown below.

-p. 5: Regarding: "Loss of DENR or MCTS1 reduces reinitiation after uORF translation in flies [35], although work in human cells suggested that they primarily affect transcripts with extremely short, single-codon uORFs [63]." The authors should consult and cite the paper by Bohlen et al (2020) (<https://doi.org/10.1038/s41467-020-18452-2>) where evidence is presented that DENR promotes reinitiation only when the penultimate uORF codons are dependent on DENR for 40S recycling. They should discuss

here or in the DISCUSSION whether the C/EBPa and C/EBPb uORFs, as well as the shortened versions of the uORFs, contain a DENR-dependent penultimate codon and the implications for the effects of depleting DENR on reinitiation following these different uORFs.

-p.6 Regarding: "Instead, this shift likely reflects reduced recognition of the long isoform start codon.", reducing eIF1 levels is expected to increase initiation selectively at poor-context start codons, so it's unclear how recognition of the long isoform AUG would be diminished owing to its Kozak context, whether optimal or otherwise. This needs to be reconsidered and clarified in the text.

-p.6 Regarding: "we speculate that extreme, CRISPR-mediated eIF2 α depletion may increase leaky scanning past the long isoform start codon as a result of reduced ternary complex formation ", they should cite literature establishing that reinitiation at a proximal downstream AUG can be suppressed at low ternary complex levels. They may also want to note that CRISPRi of two eIF2B subunits has the same effect in their data, consistent with lowered ternary complex levels being responsible.

-p.6 Regarding: "In general, ribosomes must be recycled after translation termination in order to prepare them for a new round of initiation." Only release of deacylated tRNA, not subsequent dissociation of the vacant 40S from the mRNA, promotes reinitiation, by allowing binding of a new ternary complex to the P site of the retained 40S that resumes scanning. Thus, only the first stage of 40S recycling promotes reinitiation. Again, consult/cite Bohlen et al Nature (2020).

-Fig. 5G has not been cited in RESULTS, but should be cited on p. 9.

-p.9 Regarding: "We also observed a substantial increase in long isoform expression when we extended the uORF, suggesting that this change disrupted some inhibitory effect", Could increased leaky scanning of uORF1 occur on lengthening the uORF owing to reduced queuing by scanning PICs back to the uORF start codon conferred by a stalled 80S ribosome in the uORF? Such a queue should also be diminished by inhibiting TORC1 and could explain increased bypass of the uORF on inhibition of TORC1. I believe this sort of explanation is actually advanced by the authors later in the DISCUSSION, and could be alluded to here briefly.

- Why do the results for DENR depletion differ between Fig. 6C and S6B? Actually, how do these two figures differ for both the DENR and PELO results as they seem to be identical experiments?

-It seems important to show a replicate experiment for the Western data in Fig. S7A, to show that the small reduction in long isoform is reproducible.

-It is probably worthwhile noting in the DISCUSSION that the magnitude of reduction in the long isoform for the native C/EPBa mRNA shown in Fig. S7 on PELO depletion might be too small to affect cell biology unless it is combined with mutations in other factors that would either increase the magnitude or combine it with reinforcing changes in related pathways.

We thank the reviewers for their insightful and constructive feedback on our manuscript. Through these reviews, we implemented several revisions that we believe had a strongly positive impact on the quality and clarity of our work.

Reviewer #1 (Comments to the Authors (Required)):

In this manuscript, the group of Nicholas Ingolia reports the discovery that PELO is a new player in translation reinitiation and ribosome recycling. They find PELO is key to regulating the synthesis of two alternative polypeptides during the translation of C/EBPalpha mRNA. The two isoforms derive from two different start sites. Overall, the paper moves ahead in a field hardly explored in mRNA translation initiation, namely the reinitiation process of translation and what factors are involved in it. For all conclusions, the authors show strongly supportive data. The paper is well written and the experiments are well performed. Before publication, some minor points should be addressed:

1. *Authors should explain (introduction) what is the difference, in terms of mechanism, between reinitiation and ribosome recycling.*

We agree that the relationship between recycling and reinitiation is central to our work. We have added the following explanation to the Introduction:

“Alternatively, uORF start codon recognition can lead to reinitiation wherein ribosomes translate the uORF, but then remain bound to the mRNA due to incomplete recycling after termination and reinitiate at a second, downstream ORF [25, 27, 29]. Typically, after translation termination, the ribosome is split apart and the large (60S) subunit is released [30, 31]. The P-site tRNA is then removed, and finally the small (40S) subunit is released from the mRNA as well, to complete the recycling process [32, 33]. Translation reinitiation occurs when recycling is incomplete, and either the whole ribosome or the small subunit remains bound to the mRNA and begins translation again at a different start codon.”

2. *In the Abstract, they should correct the definition of mTOR. "mTOR" stands for mammalian target of rapamycin, not "mechanistic target of rapamycin."*

While mTOR was originally discovered and named as the mammalian target of rapamycin, orthologues were later found across many eukaryotes (far beyond mammals) and so it was renamed as the “mechanistic target of rapamycin”, in HGNC and other databases.

3. *In Figure 1 or Figure 6A, the authors should include the complete Kozak sequence of the two alternative ORF, from the -3 to the +4 nucleotides surrounding both AUG start codons.*

The complete start codon context has been added for the two start sites in Figure 1. Additionally, all 5' UTR sequences used in the reporters in Figures 4 and 6 are in Supplementary Table S1.

4. *Is there evidence that misregulation of the translation of the two alternative ORFs of C/EBPalpha has actual consequences in hematopoiesis and leukemogenesis? It would be worth*

performing a survey in the reported proteomes of leukemia to see if there is evidence of unregulated synthesis of the two alternative C/EBP α polypeptides.

We note that Koschmieder et al. (2007) (reference 107) in a manuscript entitled “CDDO induces granulocytic differentiation of myeloid leukemic blasts through translational up-regulation of p42 CCAAT enhancer binding protein alpha” (where p42 is the long isoform of C/EBP α) report that, “CDDO increases the ratio of transcriptionally active p42 and the inactive p30 CEBPA isoform, which, in turn, leads to transcriptional activation of CEBPA-regulated genes (eg, GSCFR).”

Conversely, Schuster et al. (2013) (reference 109) is entitled “Lack of the p42 form of C/EBP α leads to spontaneous immortalization and lineage infidelity of committed myeloid progenitors” and describes how mutations that eliminate the long p42 form while leaving the short isoform intact are driver mutations in AML. Because *CEBPA* mutations arise in 10-15% of leukemia patients, it would be hard to distinguish dysregulated *CEBPA* translation from other mechanisms that alter the isoform ratio. Nonetheless, these mutations attest to the central importance of the isoform ratio in leukemogenesis.

We have included the following comment in the Discussion to further highlight this point:

“C/EBP α is a critical hematopoietic transcription factor whose expression is regulated at multiple levels to ensure proper differentiation and control cell fate decisions. In the myeloid lineage, increasing long isoform expression over short drives granulocytic differentiation [106, 107], whereas exclusive expression of the short isoform ratio can block this transition [108] and can promote leukemogenesis [52, 57, 109].”

Reviewer #2 (Comments to the Authors (Required)):

CEBPA is a transcription factor involved in cell differentiation of multiple tissue lineages. Its expression is regulated at different levels, including during protein synthesis. uORF-dependent control of start codon choice results in the production of alternative protein isoforms from a single CEBPA mRNA. The molecular players in the translational control of CEBPA mRNA remain unknown. As disruption of the CEBPA protein isoform ratiometric balance is associated with human cancers, elucidation of the underlying molecular mechanisms may uncover new therapeutic strategies.

In this study, the authors performed a CRISPRi screen using a dual fluorescent reporter mimicking CEBPA mRNA translational control. With this experiment, Fernandez et al. found multiple hits that caused shifted CEBPA isoform ratio. Among the factors producing the strongest and most significant changes in isoform ratio were translation factors known to regulate reinitiation (DENR, eIF3H and eIF4G2) and the ribosome rescue factor Pelota (PELO). The unexpected association of a ribosome recycling factor with uORF-mediated regulation of protein synthesis prompted the authors to understand the role of PELO in start codon choice on the CEBPA mRNA. Based on their results, the authors add a new layer in the control of CEBPA isoforms expression in cells. Following uORF translation, PELO is required to recycle and remove post-termination ribosomes obstructing the start codon required for the synthesis of the

long CEBPA isoform. Thus, in conditions of decreased PELO activity, the ratiometric balance of CEBPA isoforms is altered.

The experiments presented in the manuscript are elegant, solid and well performed. Attesting the quality of their experiments, several of the results support previous studies addressing the translational regulation of CEBPA mRNA and the role of PELO, DENR and eIF4G2 in translation. There are however, several concerns that could be addressed to further support the proposed model on the regulation of CEBPA mRNA translation by PELO.

1. The authors propose that PELO enhances long CEBPA isoform expression by recycling and removing ribosomes that were engaged in uORF translation and that block start codon recognition. However, none of the experiments showed that PELO associates with the 5' UTR of PELO mRNA, i.e., with the post-termination ribosomes following uORF translation. An experiment addressing the PELO binding to the 5' UTR of the CEBPA reporter (with and without uORF) might show the specific association of PELO with these post-termination ribosomes.

We appreciate the reviewer's suggestion and agree that we do not directly capture PELO's interaction with post-termination ribosomes in the 5' UTR of *CEBPA*. However, PELO does not bind mRNAs directly; it interacts with ribosomes, and triggers the dissociation of the ribosomes and their release from the mRNA (Pisareva et al. 2011, doi: 10.1038/emboj.2011.93, reference 58). For this reason, we would not expect to detect PELO binding to the 5' UTR of the reporter.

Furthermore, though our study does not show PELO's association with post-termination ribosomes in the 5' UTR directly, extensive prior biochemical and structural studies have established that PELO binds ribosomal complexes after termination and suppresses the effects of large subunit recycling defects (Shoemaker et al., Science 2010; Becker et al., Nature 2012; Guydosh et al., Cell 2014; Young et al., Cell 2015, references 102, 59, 60 respectively). We also believe that the dependence of PELO's effect on *CEBPA* uORF translation as demonstrated by our knockdown experiments in cell lines containing either no uORF or a uORF start codon mutant (as shown in Figure 4B) does demonstrate a genetic, though indirect, association. We have amended the text in the Discussion to include this limitation to our proposed model as follows:

“Notably, however, PELO is recruited by unrecycled ribosomes themselves and once recruited it drives their removal from mRNAs [102] and so it is not possible to capture a direct association of PELO with substrate transcripts.”

2. Wouldn't it be reasonable to assume that in PELO KD conditions, the excess of unrecycled ribosomes in the 5' UTR would increase the chances of reinitiation and therefore the synthesis of the short CEBPA isoform? However, PELO KD seems only to affect the expression of the long CEBPA isoform. Based on the ribosome profiling data, DENR and eIF4G2 levels are not significantly altered upon PELO knockdown. Can the authors comment on this observation?

The reviewer highlights a legitimate hypothesis given our data showing increased uORF translation upon *PELO* KD (Figure 5A-B). We now include commentary in the Discussion pointing to the contribution of non-AUG initiation in the 5' UTR as one potential factor explaining why short isoform levels remain similar between *PELO* KD and control conditions:

“Accumulated post-termination ribosomes could also promote near-cognate initiation at non-AUG codons upstream of the uORF, which would also reduce long isoform expression specifically.”

This question further raises an important distinction between early recycling failures, that leave a complete but vacant ribosome on the mRNA, and later recycling failures that leave only a small subunit. PELO resolves these early recycling failures, but should have no effect on later steps of recycling that appear to be required for downstream reinitiation. We now explain this distinction in the introduction and clarify throughout the manuscript and make explicit mention of it in the Introduction here:

“By facilitating small subunit recycling after uORF translation, these factors modulate the stress-inducible synthesis of the bZIP transcription factor, ATF4 [35–38]. While defects in large subunit recycling can also permit aberrant reinitiation, this is mechanistically distinct from small subunit recycling and not associated with uORF regulation [39, 40].”

We also don't expect that translation of DENR, which does not contain a uORF, to be significantly altered upon *PELO* depletion; while *eIF4G2* does contain a uORF, it is located more than 400 nucleotides upstream from the main start codon. We show in Figure 6 that extending the distance between the uORF and main ORF by just 7 or 10 nucleotides reduces the effect of *PELO* knockdown on our reporter, so it is unlikely that the *eIF4G2* uORF is affected by the same queuing mechanism as on the *CEBPA* transcript.

3. *Figure 5 and Supplementary Figure 5 - the ribosome footprint density along the CEBPA reporter mRNA in PELO gRNA cells seems to be increased not only in the vicinity of the uORF and the start codon of the long isoform, but across the 5' UTR and CDS (Supplementary Figure 5D). Is this correct? It is a bit difficult to assess the difference between control and PELO gRNA cells in Supplementary Fig. 5D (too small). Can the authors please provide some other type of figure/quantification showing the differences in ribosome footprints along the CEBPA mRNA in control and PELO gRNA conditions?*

We agree with the reviewer that the ribosome profiling data as presented in Figure 5 and Supplementary Figure 5 did not clearly convey the differences in ribosome occupancy in our *PELO* sgRNA condition. To address this, we have now added plots of ribosome density (count/length) across each region of our *CEBPA* reporter in Figure 5B as well as to Supplementary Figure 5E.

4. *If the hypothesis proposed by the authors on the function of PELO in CEBPA mRNA start codon choice is correct, the increased ribosome footprints around the long isoform start codon in PELO gRNA cells should disappear or no longer concentrate at this position in the CEBPA reporter mRNA that lacks the uORF start site, since in this condition the expression of the long isoform increases and PELO is no longer required. Can the authors show this effect on the delta uORF start reporter mRNA? Alternatively, in a previous study (Mills et al. Cell Reports 2016) the authors addressed the presence of unrecycled ribosomes in the 3' UTR by treating cell*

lysates with high salt concentrations to release vacant ribosomes. A similar treatment with the CEBPA mRNA from PELO gRNA treated cells would show that the footprints observed at the long isoform start site correspond to post-termination ribosomes.

While we agree with these predictions, the proposed experiment is beyond the scope of these revisions. It is not feasible to target ribosome profiling to specific transcripts, and so the proposed experiments would involve transcriptome-wide profiling and sequencing to address a single mutant reporter.

Minor points:

1. A detailed description of the calculation used to determine the relative abundance of the two translational isoforms by fluorescence ratio would benefit the reader and interpretation of the results. Are fluorescence absolute values used, or median values? This information could be provided in the material and methods section.

The fluorescence ratio was determined by taking the absolute (after calibration with a no fluorescence control cell line) mNeonGreen2 fluorescence values and dividing them by the absolute mScarlet fluorescence values per cell. Absolute values are depicted in histograms while median values are displayed in bar plots. A more detailed description of all ratiometric calculations has been added to the Methods under “Flow Cytometry” as follows:

“The green/red fluorescence ratio was calculated by taking the ratio of absolute mNG2 fluorescence values to the absolute mScarlet fluorescence values per cell. To determine the normalized fluorescent values in the three color reporter assays, total levels of long and short isoforms were calculated by measuring green and red fluorescence in C/EBP α reporter lines expressing only either the long or short isoforms due to mutations in either the uORF start codon to abolish short isoform expression or the long isoform start codon. These fluorescence values were then divided by iRFP levels per cell to obtain normalized green and red fluorescence levels.”

We appreciate the reviewer’s feedback on this point as it is indeed critical to the interpretation of all of our fluorescent measurements.

2. In the knocked down experiments performed in the manuscript the authors have only showed that CRISPRi gRNAs reduced PELO protein levels. Other factors were also downregulated (e.g.: DENR, eIF4G2...) but the efficiency of the transcriptional inhibition on protein levels is not shown in any of the experiments. In addition, complementation experiments were only performed for PELO. Can the authors also rescue the effects on start codon choice with DENR and eIF4G2?

We have performed additional qPCR measurements of knockdowns against *DENR* as well as another hit from our screen, *CNOT9*, in Supplementary Figure 3C. Though we did not perform complementation experiments for *DENR* and *eIF4G2*, *eIF4G2* depletion has been shown to impact *CEBPA* translation (see translation efficiency changes in Weber et al. 2022, reference 74)

and we have added additional text discussing their potential roles on the *CEBPA* uORF to the Discussion below:

“We also show that *CEBPA* uORF translation is dependent on DENR/MCTS1 and DAP5. The DENR/MCTS1 heterodimer recycles 40S subunits after termination [32, 71, 88, 105], and this activity is important for subsequent reinitiation on many transcripts, including ATF4 [34, 37, 73]. The need for DENR in post-termination 40S recycling varies according to the identity of the tRNA being evicted, and thus the penultimate codon [37]; the *CEBPA* uORF ends with a Leu codon, which is expected to confer strong DENR dependence. The *CEBPA* 5' UTR is highly structured as well, similar to other mRNAs that depend on DAP5 for efficient translation [74]. While the molecular details of DAP5 function are not clear, DAP5-dependent translation requires efficient termination and recycling, again connecting the end of uORF translation with downstream initiation.”

3. In page 8, the authors refer to other studies where *PELO* KO/KD cells/tissues have been used in conjunction with ribosome profiling. Have the authors checked the footprint densities on endogenous *CEBPA* and *CEBPB* mRNAs (in case the mRNAs were expressed in the profiled tissues), and is the footprint distribution similar to the one described in the manuscript?

We appreciate the reviewer's suggestion and subsequently checked whether C/EBP α was expressed in the mice knockout study in Terrey et al. 2021, reference 83, as well as the ribosome profiling study in human platelet and reticulocyte cells in Mills et al. 2016, reference 61 and unfortunately in both studies, *CEBPA* mRNA was either not expressed (in the human study) or expressed at very low levels (in the mouse study). In the mouse *PELO* knockout study, only 200-300 ribosome footprint reads mapped to the *CEBPA* transcript, which is orders of magnitude lower than the expression we see in our reporter cell lines (2-4000 reads), making interpretation of ribosome footprint distributions unreliable.

4. The authors have used a split, self-complementing green fluorescent protein mNeonGreen2 system to score green fluorescence. Can changes in mNG2 1-10 expression levels following CRISPRi indirectly affect the green fluorescence measured in the experiments? Since RNA-seq was performed, can the authors show that in the positive hits associated with decreased green fluorescence, mNG2 1-10 mRNA levels remain unaltered?

We agree with the reviewer that this is an important control and we have added a plot comparing levels of mNG2 (1-10) from our *PELO* KD RNA-seq experiments to Supplementary Figure 5F and show that levels of the large fragment remain unchanged between conditions.

5. The authors should refer to the supplementary tables in the text of the different sections of the manuscript.

We have added references to the supplementary tables which describe plasmid sequences, primers and sgRNAs used in our manuscript to the text in the appropriate sections.

6. The authors refer in page 11 that expression of *PELO* is decreased in AML. Can they corroborate that AML cell lines have higher levels of the short *CEBPA* isoform using western blotting analysis?

While we do not show this in our study, previous work has shown that various AML derived cell and mouse lines do indeed express an overabundance of the short C/EBP α isoform, which is often referred to in this literature as the p30 isoform (see references 52, 57, 108-109).

7. Figure 2 - how would the *mTOR* inhibitor PP442 and the delta *uAUG* results score in these bins?

The Δ uAUG cells would be sorted entirely into the “far-green” bin; it’s harder to estimate for PP242, but S1C shows some overlap between the DMSO and PP242 histograms and so we would expect a strong but incomplete shift towards the right, probably comparable to DENR in Figure 2B and 3A.

8. Fig. 2C and D - data could be better understood if the different bins (FR, NR, NG, FG) would also be indicated in the graph (dashed lines or shadows). In addition, the supplementary tables with the scores for the different gRNAs could also benefit if bin information was also present for each gene with a significant effect on isoform ratio shift (or indicate genes that enhance long or short isoform expression).

These data summarize the distribution of the guide RNA enrichment across 4 different bins and so it isn’t possible to draw the requested demarcations. To address this concern, we have added a plot of the green/red distribution of our reporter cell line transformed with one of the sgRNA libraries and the bins used for sorting to Supplementary Figure 2B. We have also added raw count information from each bin for each sgRNA across all of our libraries to the corresponding supplementary tables.

9. Fig. 2E and F – have the authors also analysed the GO terms enriched in the sgRNAs that produced a significant shift towards short isoform?

We have now added a GO analysis of the terms associated with sgRNAs that shifted the fluorescence ratio towards the short isoform for the Cancer and Apoptosis library shown in Figure 2D and 2F to Supplementary Figure 2 (Figure S2F). There were no significant GO terms for the equivalent sgRNAs in the Gene Expression library shown in Figure 2C and 2E.

10. A western blotting analysis of *CEBPA* isoforms in *PELO* KD conditions (Fig. 3) would allow the reader to understand the shift in isoform expression. How much of long isoform is still produced?

We quantify absolute abundance of these isoforms under different knock-down conditions by fluorescence in Fig. 4 and Supplementary Figure 4F. Specifically, *PELO* KD decreases long isoform expression by ~30% relative to control, a less than two-fold difference, and fluorescence provides much more robust quantitation than blotting.

11. Fig. 4D and E would benefit the interpretation of the results if the % of long and short isoforms in total CEBPA protein was also displayed in the graphs. The total normalized fluorescence levels change between conditions making it difficult for the reader to compare all the conditions.

We now include these ratios in the graph.

12. Fig. 5B - The information of the numbers of genes present in each quadrant could be presented.

The number of genes in each quadrant have now been added to the scatterplot in Figure 5C (formerly Figure 5B).

13. Fig. 6B, D and F - quantification (bar graph) of the effect of uORF size, spacer and 5'UTR in control cells would help the reader to understand the effect of each parameter in the green/red ratio.

We have added quantification of these effects under control conditions (nontargeting sgRNA) in Figure 6 and provide further analysis of how these reporters change long and short isoforms levels in Supplementary Figures 6A, B and D.

14. Sup Fig. 1 - x axis scale is missing. Is overall translation also inhibited? Overall red fluorescence decreased or not? What is the ratio of green in the total red? Or, is only the ratio of green to red that it is altered?

We apologize for that oversight and have amended Supplementary Figure 1 to include axis labels. We provide further quantification of the effect of PP242 in Figure 5G-H.

15. Sup Fig. 4B - lane 9 of WB is more similar to lanes 10-12 than to lanes 7-8. Is the blot mislabelled?

We believe that differing levels of proteolysis affected overexpressed PELO and to a lesser extent HaloTag, explaining the presence of V5-tagged degradation products in this blot. These effects may occur during sample preparation, or in cells through the chemically regulated degron used to control PELO and HaloTag expression.

16. Sup Fig. 4 - a short explanation of the shld 1 effect on PELO would benefit the reader (for instance in the figure legend or material and methods).

A more detailed description of the Shield1 experiments have been added to the Results section when first introduced:

“We next devised a complementation system to rescue PELO depletion by fusing a drug-controlled destabilization domain (DD) that is stabilized in the presence of the small molecule, Shield1 [89], to the PELO and HaloTag CDS (Figure S4A and S4B).”

As well as to the Methods under “PELO re-expression rescue assays” as follows:

“The HaloTag and PELO CDS was fused to the FKBP12 destabilizing domain (DD) to generate (N-terminal) DD-HaloTag and PELO-DD (C-terminal) constructs. DD-containing constructs are stabilized by binding to the small molecule, Shield1 [89], such that in the absence of Shield1, the resultant proteins are rapidly degraded.”

17. *Protein size ladder missing in many of the WBs (kDa)*

Ladders have been added to the appropriate Western blot analyses.

18. *Sup Fig. 7 - This result would be easier to analyse if the protein bands were better separated. It looks like two long isoform bands are present and that only of these is reduced following PELO knock down. Can the authors repeat the blot with a gel that runs longer or of different acrylamide percentage to better discriminate the different CEBPA isoforms expressed in these cells? In addition, a scheme describing the different start sites originating the multiple CEBPA isoforms in THP1 cells would be helpful.*

A second, long C/EBP α isoform is indeed expressed from endogenous *CEBPA* transcripts from an AUG start codon that is located 14 codons downstream from the main long isoform. This second long isoform is considered functionally equivalent to the main long isoform. This has been clarified in a schematic of the protein structure of endogenous C/EBP α and has been added to Supplementary Figure 7A.

Reviewer #3 (Comments to the Authors (Required)):

This is an interesting paper that uses a very creative forward selection involving a 2-color reporter and FACS analysis to screen CRISPRi libraries in mammalian cells to identify trans-acting factors affecting uORF-mediated translational control of the *C/EBPa* gene. They identify DENR and DAP5 and provide evidence from reporter experiments that they act by enhancing reinitiation following translation of the uORF, which leads to increased initiation further downstream from the next AUG codon initiating the long *C/EBPa* isoform at the subsequent AUG codon for the short isoform. These results are consistent with previous findings on these two factors in promoting reinitiation after uORF translation; although in the case of DENR, the stimulatory effect has been found to be highly dependent on the last, penultimate codon of the uORF, which the authors did not address. Importantly, they also implicate for the first time the ribosome dissociation factor PELO in controlling the effects of a regulatory uORF in blocking initiation at a downstream CDS start codon. Their reporter data indicate that depleting PELO reduces expression of the *C/EBPa* long isoform without impacting reinitiation at the downstream short isoform AUG codon, suggesting (at least to this reviewer) that leaky scanning through the uORF AUG codon, which is necessary to initiate at the second AUG for the long isoform, is impaired. Ribosome profiling data of the native gene on PELO depletion shows increased RPF occupancies in the uORF, which would fit with increased uORF translation and decreased leaky scanning to the long isoform AUG; however, it also gives the unexpected result of increased 80S ribosome occupancies at the long isoform AUG codon, at odds with the conclusion from reporter data that initiation at the long isoform AUG is reduced. They attempt to explain this by

proposing that the increased RPFs at the long uORF AUG codon represent vacant 80Ss that escaped recycling by PELO after terminating at the uORF stop codon and that they interfere with initiation by the 40S preinitiation complexes (PICs) that leaky-scan through the uORF AUG codon. They attempt to support this mechanism by showing that lengthening or shortening the uORF, or increasing the separation between the uORF stop codon and long isoform start codon diminish the impact of depleting PELO on long isoform initiation; however, it's not clear to this reviewer how these findings would support their model. They present western blot evidence that PELO depletion reduces expression of the native long isoform of C/EBPa mRNA, and also reporter data indicating that PELO functions in the same manner for the similarly regulated C/EBPb mRNA. Consistent with previous data, depletion of PELO activates mTOR, which was shown previously to decrease the ratio of long:short C/EBPb isoforms; however, they show that PELO depletion reduces long isoform translation independently of its activation of mTOR by showing that the reduction in long isoform translation conferred by PELO remains evident after depletion of mTOR or RHEB.

The results implicating PELO for the first time in controlling translational control by the uORFs in the C/EBPa and b mRNAs in a manner that specifically reduces translation of the long isoforms are convincing and represent an important accomplishment.

*1. However, I am not convinced of the proposed mechanism. The retention of vacant 80S ribosomes at the uORF stop codon that would physically obscure the AUG codon of the long isoform is not supported by the profiling data, **which show no 80S peak at the uORF stop codon**. Instead, the elevated 80S peak is located at the AUG of the long isoform. It's possible, as they allude to, that these are vacant 80S complexes that have migrated from the uORF stop codon to the AUG of the long isoform, but this does not seem to be their model based on its description in the DISCUSSION where they indicate that an 80S ribosome that is not released from the uORF stop codon after uORF translation would directly block access to the long isoform start codon because of the close spacing between the stop and start codons. The other problem is that if a vacant 80S is blocking initiation at the long isoform AUG codon by any 40S PICs that leaky-scanned the uORF then it should also block progression of these scanning PICs to the next AUG for the short isoform and thereby reduce reinitiation at the latter start site.*

Regarding the position of the unrecycled 80S ribosomes, we entirely agree that post-termination ribosomes can migrate. For instance, Skabkin et al., 2013 note in their abstract that, “Strikingly, post-termination ribosomes were not stably anchored on mRNA, and migrated bidirectionally...” Likewise, when Rli1/ABCE1 are depleted in living cells, vacant ribosomes are distributed across the 3' UTR.

We have adjusted the text in the Discussion to clarify our meaning:

“an 80S ribosome that is not recycled off of the mRNA after uORF translation would directly block the long isoform start codon” (emphasis added).

More broadly, we would propose that each ribosome undergoes one of a few fates, determined by a series of stochastic events. Some ribosomes do not undergo early ABCE1-mediated 60S recycling and remain as inhibitory vacant 80S ribosomes; notably, early recycling failure is not

associated with reinitiation after uORF translation. Other ribosomes do undergo 60S recycling, followed by DENR/MCTS1-mediated tRNA removal, and are competent to reinitiate downstream. Finally, we presume that some ribosomes succeed at all stages of recycling and are released after uORF translation. Reinitiation may also be impaired by post-termination ribosomes, and the state of the mRNA (i.e., the presence or absence of a vacant 80S ribosome) is another stochastic factor.

The isoform ratio is determined by the relative frequencies of these different outcomes, averaged over many translation initiation events. The steps of recycling form an ordered pathway, and the likelihood of success at each step is probably affected at least in part by the abundance of the factors (DENR/MCTS1, perhaps others) that drive it.

Finally, we would note that a 40S PIC reaching the long-isoform start codon by leaky scanning will encounter any post-termination ribosome in that region as a scanning PIC. In contrast, a ribosome translating the uORF will encounter such a post-termination ribosome while translating the uORF as an 80S ribosome (keeping in mind the ~30 nt width of each ribosome), likely displacing the post-termination ribosome. A translating ribosome is much better able to move through impediments than a scanning 40S PIC. This interaction, which happens while the ribosome is translating the uORF, means that a ribosome finishing uORF translation will encounter a different context than a 40S PIC bypassing the uORF. While we do not know the details of these scenarios, they are likely different, and this difference explains the different impact of post-termination ribosomes on leaky scanning to the long isoform versus uORF translation and reinitiation.

2. I also don't understand how their model is supported by the observation that the impact of PELO depletion is reduced by changing the uORF length, as these changes should not alter the retention of unrecycled 80Ss at the uORF stop codon.

We reason that the positioning between a post-termination ribosome located either at the uORF stop codon or at the long isoform start codon and the uORF start codon (which would be approximately a single ribosome footprint's distance in the wildtype *CEBPA* uORF) is important for queuing-mediated start codon recognition. Changing the length of the uORF consequently alters this positioning thereby reducing the impact of ribosome queuing on uORF start codon recognition. The reviewer is correct, however, that this should not impact the inhibitory effect of these unrecycled ribosomes on long isoform expression and we note that changing the length of the uORF does not have as strong of an effect on the PELO-mediated phenotype as extending the intercistronic spacer length (Figure 6C vs 6E).

3. And while I follow that increasing the spacing between the uORF stop codon and AUG for the long isoform from 7 nt to 14nt or 17nt would allow a PIC to form at the long isoform AUG even if an unrecycled 80S is still positioned at the uORF start codon, I don't see how the scanning PIC can make it through the unrecycled 80S to benefit from the expanded separation.

We now clarify that the effect of uORF length and position may affect collisions and queuing of uORF ribosomes, which has been reported on other transcripts. We describe this in the Introduction, "This kind of ribosome queuing has also been shown to increase recognition of

upstream start codons,” and in the Discussion, “Additionally, this post-termination ribosome could stall scanning pre-initiation complexes near the uORF start codon, enhancing uORF translation....Similar enhancements of upstream initiation are seen in other situations when scanning is blocked”, citing Ivanov et al. 2018, Manjunath et al. 2019, and Vindu et al. 2021 as well as our own previous work.

4. The authors might consider instead that PELO depletion reduces leaky scanning of the uORF AUG codon to reduce the number of ribosomes able to scan past the uORF to the long isoform AUG, but in a manner that does not lead to increased reinitiation at the short isoform AUG. Increased initiation at the uORF AUG is supported by the profiling data, which also suggests increased initiation at near cognate start sites upstream of the uORF. Perhaps, retention of an unrecycled 80S at the stop codon on PELO depletion leads to a translation pause in the uORF that slows down scanning PICs approaching the uORF. This leads them to initiate aberrantly at near-cognates upstream of the uORF, and also to increase their probability of recognizing the uORF AUG to thereby reduce leaky scanning to the long isoform AUG. (The authors have already mentioned this type of queuing in the uORF that would reinforce initiation there.) If one proposes that the net outcome of slower elongation and increased initiation in the uORF yields essentially the same number of 80S ribosomes that complete uORF translation and go on to reinitiate at the short isoform AUG, this would lead to no reduction in short isoform initiation and explain the observed specific reduction in long isoform translation. If the unrecycled 80S tends to migrate away from the uORF stop codon and stall at the long isoform AUG without completing initiation, and then moves on or falls off the mRNA, this would explain their profiling data.

We do agree that our model is inferred from indirect genetic data that are consistent with alternative models as well. Because we saw weak or no effect from start codon stringency factors EIF1 and EIF5, relative to stronger effects seen from recycling and rescue factors, we had not proposed a central role for non-AUG initiation in our model – though we do see increased ribosome density in the 5'UTR upon *PELO* depletion (Figure 5B). Nonetheless, non-AUG initiation does occur and we appreciate this suggestion, which we have added to the Discussion:

“Accumulated post-termination ribosomes could also promote near-cognate initiation at non-AUG codons upstream of the uORF, which would also reduce long isoform expression specifically.”

We have also added this point to the model schematic in Figure 7.

The other problem is that if a vacant 80S is blocking initiation at the long isoform AUG codon by any 40S PICs that leaky-scanned the uORF then it should also block progression of these scanning PICs to the next AUG for the short isoform and thereby reduce reinitiation at the latter start site.

We believe this is a duplicate of the last sentence in point 1, above, which is addressed in our response to that comment.

5. Two other issues with the profiling data are that we are not being shown replicates, and we are not being shown the region of the main CDS up to and beyond the AUG for the short isoform to evaluate whether the expected reduction in RPF density between the AUGs for the long and short isoforms is actually evident on *PELO* depletion. If it is not, then the utility of the profiling data for revealing the mechanism is questionable.

We have now added ribosome occupancy profiles of each replicate to Supplementary Figure 5 (Figure S5D) and have also included quantification of the footprint distribution across the long isoform only region as well as the shared region between the two isoforms (Figure S5E). This analysis shows that knockdown of *PELO* results in a decrease in overall C/EBP α translation in line with our reporter assay results yet does not clearly show a depletion in ribosome density in the region between the long and short isoform start codons (displayed as the “long isoform region”). We propose that these footprints include unrecycled ribosomes that accumulate upon *PELO* depletion—similar to the accumulation of ribosomes across 3' UTRs—and have described this effect as follows:

“Indeed, quantification of ribosome footprint density revealed a slight increase across just the long isoform region despite the overall decrease in translation of the reporter suggesting the presence of these unrecycled ribosomes (Figure S5E).”

6. Fig. 1D: it is expected that deletion of the uORF start will increase long isoform production, but this blot seems to show reduced long isoform instead, unless this is a loading problem. It's shown later in Fig. S6D, that this predicted effect is observed on removing the uORF for C/EBPb for the normalized reporter. I suggest they include a similar figure for C/EBP α and note in the text that the expected result will be shown below.

We show in Figure 4C that removing the uORF start codon causes a switch to exclusively long isoform production with no change in total protein produced.

7. p. 5: Regarding: "Loss of *DENR* or *MCTS1* reduces reinitiation after uORF translation in flies [35], although work in human cells suggested that they primarily affect transcripts with extremely short, single-codon uORFs [63]." The authors should consult and cite the paper by Bohlen et al (2020) (<https://doi.org/10.1038/s41467-020-18452-2>) where evidence is presented that *DENR* promotes reinitiation only when the penultimate uORF codons are dependent on *DENR* for 40S recycling. They should discuss here or in the DISCUSSION whether the C/EBP α and C/EBPb uORFs, as well as the shortened versions of the uORFs, contain a *DENR*-dependent penultimate codon and the implications for the effects of depleting *DENR* on reinitiation following these different uORFs.

We agree with the reviewer that this is an important point that should be discussed have addressed this in the Discussion as follows:

“The need for *DENR* in post-termination 40S recycling varies according to the identity of the tRNA being evicted, and thus the penultimate codon (Bohlen et al., 2020); the *CEBPA* uORF ends with a Leu codon, which is expected to confer strong *DENR* dependence.”

8. p.6 Regarding: *"Instead, this shift likely reflects reduced recognition of the long isoform start codon.", reducing eIF1 levels is expected to increase initiation selectively at poor-context start codons, so it's unclear how recognition of the long isoform AUG would be diminished owing to its Kozak context, whether optimal or otherwise. This needs to be reconsidered and clarified in the text.*

We agree that the original version did not capture our meaning well and we have revised the text to read, "Instead, this shift likely reflects other, potentially indirect, effects on the isoform ratio."

9. p.6 Regarding: *"we speculate that extreme, CRISPR-mediated eIF2 α depletion may increase leaky scanning past the long isoform start codon as a result of reduced ternary complex formation ", they should cite literature establishing that reinitiation at a proximal downstream AUG can be suppressed at low ternary complex levels. They may also want to note that CRISPRi of two eIF2B subunits has the same effect in their data, consistent with lowered ternary complex levels being responsible.*

We appreciate these suggestions, which we have incorporated in the Results by saying, "...reduced ternary complex formation (Hinnebusch et al., 2016). Consistent with this model, we also see that depleting subunits of eIF2B, which is required to regenerate ternary complex, cause a similar shift (Figure S2G)."

10. p.6 Regarding: *"In general, ribosomes must be recycled after translation termination in order to prepare them for a new round of initiation." Only release of deacylated tRNA, not subsequent dissociation of the vacant 40S from the mRNA, promotes reinitiation, by allowing binding of a new ternary complex to the P site of the retained 40S that resumes scanning. Thus, only the first stage of 40S recycling promotes reinitiation. Again, consult/cite Bohlen et al Nature (2020).*

We have amended the text in the Results as follows to clarify the mechanisms involved in reinitiation:

"Short isoform expression depends primarily on downstream reinitiation after uORF translation. In general, deacylated tRNAs must be released from 40S ribosomal subunits following termination in order to prepare them for a new round of initiation. The DENR/MCTS1 heterodimer promotes recycling of 40S ribosomal subunits by removing tRNAs after termination thereby allowing the binding of new ternary complex and the resumption of scanning. By resetting 40S subunits, they could favor reinitiation after uORF translation [32, 37, 71, 87, 88], explaining the effects of DENR and MCTS1 depletion on CEBPA start site selection."

11. Fig. 5G has not been cited in RESULTS, but should be cited on p. 9.

We have added the proper citation to the Results section as follows:

“Conversely, PELO knockdown reduced the long isoform fraction in the context of PP242 treatment, relative to a nontargeting control in PP242, consistent with a direct, mTOR-independent role for PELO in start site selection (Figure 5G).”

12. p.9 Regarding: "We also observed a substantial increase in long isoform expression when we extended the uORF, suggesting that this change disrupted some inhibitory effect", Could increased leaky scanning of uORF1 occur on lengthening the uORF owing to reduced queuing by scanning PICs back to the uORF start codon conferred by a stalled 80S ribosome in the uORF? Such a queue should also be diminished by inhibiting TORC1 and could explain increased bypass of the uORF on inhibition of TORC1. I believe this sort of explanation is actually advanced by the authors later in the DISCUSSION, and could be alluded to here briefly.

We agree with the reviewer that one possible interpretation of the increase in long isoform expression that we see upon extending the uORF could be due to increased leaky scanning past the uORF start codon. Our present data, however, does not seem to support the conclusion that this is due to reduced queuing – we believe that our data argues that we only observe queuing upon *PELO* depletion and not under wildtype conditions. It is instead possible that we have reduced the effect of an mRNA secondary structural feature important for uORF recognition by extending the uORF, given that the start codon context of the uORF variant reporters are also identical.

13. Why do the results for *DENR* depletion differ between Fig. 6C and S6B? Actually, how do these two figures differ for both the *DENR* and *PELO* results as they seem to be identical experiments?

Supplementary Figure S6B is an independent replicate of Figure 6C that also includes an assessment of the effect of DAP5 knockdown on our variant uORF reporters. We would note that the *PELO* knockdown results do not differ: the 18 nt uORF has the strongest reduction, followed by the 9 nt uORF, with the 30 nt uORF showing the weakest effect, and results are significant in both cases.

14. It seems important to show a replicate experiment for the Western data in Fig. S7A, to show that the small reduction in long isoform is reproducible.

We have encountered technical challenges in deriving THP-1 cell lines expressing CRISPRi effectors and guide RNAs; the approaches we routinely use in K562 cells have not yielded viable populations for these experiments.

We believe our data in K562 cells show that our reporter captures alternative start site regulation and enables genetic analysis of start site choice as a regulatory phenomenon, which is the focus of our work.

15. It is probably worthwhile noting in the DISCUSSION that the magnitude of reduction in the long isoform for the native *C/EPBa* mRNA shown in Fig. S7 on *PELO* depletion might be too small to affect cell biology unless it is combined with mutations in other factors that would either increase the magnitude or combine it with reinforcing changes in related pathways.

We agree with this point given the magnitude of the effect we observe in Figure S7 and have included additional text to address this in the Discussion:

“Although the isoform ratio changes we observe after *PELO* depletion are more modest, they may affect these cell fate decisions if they are reinforced by coordinated regulation in related pathways or accompanied by sensitizing mutations.”

March 27, 2024

RE: Life Science Alliance Manuscript #LSA-2023-02501-TR

Prof. Nicholas Ingolia
University of California, Berkeley
Molecular and Cell Biology
1 Barker Hall # 3202
Berkeley, CA 94720

Dear Dr. Ingolia,

Thank you for submitting your revised manuscript entitled "Ribosome rescue factor PELOTA modulates translation start site choice and protein isoforms of C/EBP α ". We would be happy to publish your paper in Life Science Alliance pending final revisions necessary to meet our formatting guidelines.

- please be sure that the authorship listing and order is correct
- please upload your main manuscript text as an editable doc file
- please upload all figure files as individual ones, including the supplementary figure files; all figure legends should only appear in the main manuscript file
- please add Keywords for your manuscript to our system
- please add the Twitter handle of your host institute/organization as well as your own or/and one of the authors in our system
- please note that the titles in the system and the manuscript file must match
- please add your main, supplementary figure, and table legends to the main manuscript text after the references section
- we encourage you to revise the figure legends for figure S2 such that the figure panels are introduced in an alphabetical order
- you may want to consider uploading Figure 7 as a Graphical Abstract rather than as a figure, but this is up to you

FIGURE CHECKS:

- please add sizes next to the blots in Figures 3C, 5F, S5I and J and S7A

A. FINAL FILES:

B. MANUSCRIPT ORGANIZATION AND FORMATTING:

Sincerely,

April 16, 2024

RE: Life Science Alliance Manuscript #LSA-2023-02501-TRR

Prof. Nicholas Ingolia
University of California, Berkeley
Molecular and Cell Biology
1 Barker Hall # 3202
Berkeley, CA 94720

Dear Dr. Ingolia,

Thank you for submitting your Research Article entitled "Ribosome rescue factor PELOTA modulates translation start site choice for C/EBP α protein isoforms". It is a pleasure to let you know that your manuscript is now accepted for publication in Life Science Alliance. Congratulations on this interesting work.

DISTRIBUTION OF MATERIALS:

Again, congratulations on a very nice paper. I hope you found the review process to be constructive and are pleased with how the manuscript was handled editorially. We look forward to future exciting submissions from your lab.

Sincerely,
